# The *Meq* Genes of Nigerian Marek’s Disease Virus (MDV) Field Isolates Contain Mutations Common to Both European and US High Virulence Strains

**DOI:** 10.3390/v17010056

**Published:** 2024-12-31

**Authors:** Joseph N. Patria, Luka Jwander, Ifeoma Mbachu, Levi Parcells, Brian Ladman, Jakob Trimpert, Benedikt B. Kaufer, Phaedra Tavlarides-Hontz, Mark S. Parcells

**Affiliations:** 1Department of Biological Sciences, University of Delaware, Newark, DE 19716, USA; jpatria@udel.edu; 2Central Diagnostic Laboratory, National Veterinary Research Institute, Vom 930101, Nigeria; jwanderluka@gmail.com; 3Department of Biological Sciences, Lincoln University, Lincoln University, PA 19352, USA; ifeomambachu6@gmail.com; 4Department of Animal and Food Sciences, University of Delaware, Newark, DE 19716, USA; vparcell@udel.edu (L.P.); bladman@udel.edu (B.L.); phaedra@udel.edu (P.T.-H.); 5Institut für Virologie, Freie Universität Berlin, Robert von Ostertag-Straße 7-13, 14163 Berlin, Germany; jakob.trimpert@fu-berlin.de (J.T.); benedikt.kaufer@fu-berlin.de (B.B.K.); 6Veterinary Centre for Resistance Research (TZR), Freie Universität Berlin, 14163 Berlin, Germany

**Keywords:** Marek’s disease, Marek’s disease virus, *meq* oncogene, virulence, virulence evolution

## Abstract

Background: Marek’s disease (MD) is a pathology affecting chickens caused by Marek’s disease virus (MDV), an acute transforming alphaherpesvirus of the genus *Mardivirus*. MD is characterized by paralysis, immune suppression, and the rapid formation of T-cell (primarily CD4+) lymphomas. Over the last 50 years, losses due to MDV infection have been controlled worldwide through vaccination; however, these live-attenuated vaccines are non-sterilizing and potentially contributed to the virulence evolution of MDV field strains. Mutations common to field strains that can overcome vaccine protection were identified in the C-terminal proline-rich repeats of the oncoprotein Meq (Marek’s *Eco*RI-Q-encoded protein). These mutations in *meq* have been found to be distinct to their region of origin, with high virulence strains obtained in Europe differing from those having evolved in the US. The present work reports on *meq* mutations identified in MDV field strains in Nigeria, arising at farms employing different vaccination practices. Materials and Methods: DNA was isolated from FTA cards obtained at 12 farms affected by increased MD in the Plateau State, Nigeria. These sequences included partial whole genomes as well as targeted sequences of the *meq* oncogenes from these strains. Several of the *meq* genes were cloned for expression and their localization ability to interact with the chicken NF-IL3 protein, a putative Meq dimerization partner, were assessed. Results: Sequence analysis of the *meq* genes from these Nigerian field strains revealed an RB1B-like lineage co-circulating with a European Polen5-like lineage, as well as recombinants harboring a combination of these mutations. In a number of these isolates, Meq mutations accumulated in both N-terminal and C-terminal domains. Discussion: Our data, suggest a direct effect of the vaccine strategy on the selection of Meq mutations. Moreover, we posit the evolution of the next higher level of virulence MDVs, a very virulent plus plus pathotype (vv++).

## 1. Introduction

Marek’s disease (MD) is a lymphoproliferative disease of viral etiology that progresses rapidly in domestic chickens (*Gallus gallus*). Despite a spectrum of disease severity, the main signs of MD are the onset of neurological syndromes, lymphoid atrophy resulting in immune suppression, skin leukosis, and the development of T-cell lymphomas in visceral organs. The latter culminates in fatality in unvaccinated, susceptible chickens within several weeks post-infection [1].

MD is caused by Marek’s disease virus (MDV), an *Alphaherpesvirinae* family member of the genus *Mardivirus*. To control MD, poultry producers worldwide employ vaccination programs composed of three genotypes of live vaccines: the attenuated MDV-1, (Gallid alphaherpesvirus 2, GaHV2), strain CVI988/Rispens, the apathogenic turkey herpesvirus (HVT, aka, Meleagrid alphaherpesvirus 1), as a monovalent vaccine, and as bivalent vaccines paired with either MDV-2 (Gallid alphaherpesvirus 3, GaHV3) strain SB-1, (HVT/SB-1) for broilers, or with CVI988/Rispens (HVT/CVI988) for longer-lived chickens (layers, broiler-breeders). Vaccine-induced immunity provides life-long protection against lymphomagenesis. Despite this control, MD immunization neither prevents superinfection nor transmission of field strains. Thus, MDV subsists within the reservoir host after exposure and infectious virus is shed with the dander into the environment. MD vaccines, albeit non-sterilizing “imperfect” vaccines, prolong the host lifespan and the shedding period in which viruses undergo continued selection [2,3]. Consequently, MDVs mutate continually into emergent field strains of higher virulence [4,5].

Presently, MDV-1 field strains are classified into four pathotypes based on the following scheme of increasing virulence: mild (m), virulent (v), very virulent (vv), and very virulent plus (vv+). The introduction of non-sterilizing MD vaccines with increased efficacy to counter vaccine-resistant pathotypes has been hypothesized to contribute to MDV virulence evolution [4,6]. This concept is based on copious surveillance of emerging field strains with respect to pathotypic classification, vaccine resistance, and year of isolation in the USA [4,5]. On an international scale, however, individual circumstances of MD prevalence and the emergence of vaccine-resistant strains in each country necessitated distinct introduction of these vaccines.

Contemporaneously with the introduction of HPRS-16 in Europe during the 1970s [7,8], the US poultry industry employed vaccination programs based on the HVT strain FC126 [9]. Both vaccines conferred effective protection against circulating vMDV strains in commercial flocks; however, Europe adopted the HVT vaccine as a surrogate to HPRS-16 shortly after licensure by the USDA in 1971 [10,11,12]. Meanwhile, the isolation of the attenuated CVI988 strain allowed for the development of the second MDV-1 serotype vaccine, which became available in Europe in 1972 [13,14]. Thereafter, CVI988 was the preferred vaccine in Europe, where protection against vvMDV strains is conferred by administering as a monovalent or as a component of polyvalent vaccines and revaccination programs [15,16].

Similarly, the monovalent HVT vaccine was used briefly in Japan, but vaccination programs have since changed following the introduction of CVI988 in 1985 and CVI988+HVT bivalent vaccines in 1988 [17]. In China, commercially available monovalent HVT and bivalent HVT+SB-1 vaccines were introduced in the 1980s, followed by the later introduction of serotype-1 vaccines. However, before the widespread availability of CVI988, the isolation of the avirulent strain 814 in 1986, which shares a common ancestry with CVI988, offered immediate protection [18].

Immunization with CVI988 and 814 monovalent vaccines or CVI988+HVT bivalent vaccines are currently used in commercial layers and breeder flocks, while commercial meat-type birds are vaccinated with monovalent HVT [19]. Conversely, the USA continued the usage of HVT as a monovalent vaccine to curtail MD losses for about a decade longer before a reduction in effectiveness became apparent [20,21]. This led to incorporating the HVT+SB-1 bivalent vaccine into US vaccination programs in 1985, which provided enhanced protection against emerging vvMDV strains [22,23]. Vaccine-resistant field strains became prevalent again in the 1990s, and eventually, the approval of CVI988 entailed effective control measures against vv+MDV strains circulating throughout US poultry flocks [24,25]. Until the 21st century, MDV field strains have demonstrated a stepwise increase in virulence each decade since the 1960s [4]. Reports of CVI988 vaccine failures, the emergence of vaccine-resistant vv+MDV field strains, and CVI988 recombination with field strains in recent years need to be addressed [19,26,27,28,29,30,31,32,33].

The viral oncogene *meq* has a central role in the pathogenesis of MDV infection, in which the gene encodes a 339 amino acid (aa) basic leucine zipper (bZIP) transcription factor (TF) in most virulent US strains, although due to variations in the number of proline-rich repeats (PRRs), 398/399 aa isoforms are found in lower virulence (BC-1, JM16) and mixed isoforms in CVI988 vaccine stocks [34]. *Meq* not only has a causal role in oncogenicity [35] mediated through bZIP interactions including the dimerization with the putative binding partner, NFIL3 as shown by Reinke et al. [36], but our group and others have identified virulence-associated mutations in the *meq* gene in a pathotype and regionally-dependent manner, but our group and others have identified virulence-associated mutations in the *meq* gene in a pathotype and regionally-dependent manner [34,37,38,39]. Based on these observations, the *meq* gene appears to undergo strong positive selection pressures, with specific mutations mapping to the C-terminal domain (CTD). It remains unclear, however, how these mutations in *meq* provide a mechanistic basis for changes in virulence and vaccine-induced immune evasion.

In Nigeria, MDV has been in circulation as early as the 1960s [40,41,42]. Since the expansion of the Nigerian poultry industry in the 1980s, recurring outbreaks of MD have been documented, and the prevalence of MD has increased with limited awareness, resources, and infrastructure to ameliorate losses [43,44,45,46,47,48,49,50].

Losses due to MD have impacted poultry operations on rural, semi-commercial, and commercial scales and can have grave economic consequences for developing countries structured on an agro-economic system. Vaccination is seldom used by semi-commercial and backyard poultry operations, upon which much of the population relies for their livelihood [44]. Vaccine procurement is difficult, and improper vaccination techniques have led to vaccine failure in recent outbreaks [51,52]. In flocks of commercially improved breeds under intensive management, Nigeria’s poultry farmers vaccinate layers due to the high risk of MD, while broilers typically are not vaccinated due to their shorter lifespan [27,43,46,47,48,50,51,53,54,55,56].

Without standardized MD vaccination programs in effect, the vaccination history of day-old chicks prior to arrival at farm sites is often unknown, leading to the common practice of farmers revaccinating flocks with HVT upon arrival. In recent years, severe outbreaks of MD in vaccinated layer or unvaccinated broiler flocks indicate that highly virulent MDVs circulating in Nigeria can break through HVT or HVT + CVI988 vaccine protection [27].

In our present study, we performed comparative sequence analysis on the complete *meq* gene sequences and partial whole genomes of field isolates from MD outbreaks in 12 farms of the Plateau State that occurred between 2015 and 2016. These are the first MDV genome sequences reported from Western Africa, and the mutations in the *meq* oncogene presented herein can elucidate MDV lineages in Africa.

The *meq* gene has been extensively studied in recent phylogenetic analyses from North America [38]; Europe: Italy [57,58] and Poland [59,60]; Asia: China [30,32,61,62,63], Japan [17,64,65], Turkey [66], India [67,68,69,70,71], Iran [72,73,74]; and Africa (Egypt) [75,76] which define the pathogenic and geographic relationships across the MDV phylogeny. The aim of this study was to use sequence comparisons and molecular phylogenetic analyses to make a parsimonious inference on MDV pathogenicity and estimate the current virulence level of circulating field strains in this disease outbreak. We report that emerging Nigerian field strains of a highly virulent MDV encoding a novel *meq* isoform was the cause of a severe outbreak of acute MD in CVI988 and subsequent HVT-vaccinated commercial layer flocks. The combinations of mutations suggest that vaccine practices could potentially lead to the emergence of vv++MDV strains.

## 2. Materials and Methods

### 2.1. Study Area

Located within Western Africa, the Plateau State in Nigeria lies in the east-central proximity of the North Central geopolitical zone, adjacent to the borders of Nigeria’s North-Western and North-Eastern zones. The area comprises commercial, semi-commercial, and small-holder backyard-scale poultry operations with moderate to high, minimal to low, and minimal biosecurity, respectively. Most of the commercial sector is based in southern states, while less than 10% are in northern zones of Nigeria. A current estimate of the poultry population and number of farms in Plateau state is difficult to ascertain, but recent studies conducted in the region that survey a sample population can serve as a proxy for this information [52,77]. Documented cases of poultry diseases have increased over the years, including the prevalence of MD in this region [44].

### 2.2. Case History

Poultry farms in Plateau State experiencing MD outbreaks were reported between 2015 and 2016. Cases of MD were severe, affecting flocks from 12 commercial poultry farm sites. The exact location of these sites and the number of chickens were unavailable; however, each farm site, the flock type, and vaccination history are specified in Table 1. Apart from one flock of backyard indigenous broiler breeds, the production type of the other 11 flocks consisted of commercially improved broilers and layers. The genetic lines and ages of the affected flocks were unavailable. MD-associated tumor incidence or clinical signs were identified in six unvaccinated broiler flocks and six vaccinated layer flocks. The layers were vaccinated at hatch (1 day-of-age) with the commercial vaccine CVI988/Rispens. The vaccination of layers was performed at the hatchery, and day-old chicks were subsequently placed upon arrival at the farms. Of the CVI988-vaccinated layer flocks, three flocks were revaccinated with HVT at 21 days-of-age. The HVT vaccine was administered at the farms by local veterinarians.

### 2.3. Clinical Samples and Pathological Findings

All farms reported increased disease incidence and mortality in unvaccinated as well as vaccinated flocks. Clinical signs are given for each flock in Table 1 and include: stunted growth, emaciation, and ruffled feathers. On postmortem examination, MD-associated gross pathological lesions were found, such as the accumulation of sanguineous fluid within the peritoneal cavity, hepatomegaly, and enlargement of the spleen. Visceral lymphomas were found in the heart, liver, and spleen.

In total, 40 samples were collected from the 12 affected poultry farms. The tissue samples for this study were collected during necropsy of infected chickens with macroscopic tumor lesions present on the liver, spleen, heart, lung, kidney, and ovary. Blood and feather pulp were also sampled from chickens, which presented clinical signs of MDV infection. Freshly collected solid tumor, blood, and feather pulp samples were smeared/spotted directly onto the active area of Whatman FTA filter paper cards (GE Healthcare, UK) and allowed to dry at ambient temperature. For molecular detection and virus isolation, FTA sample cards were shipped to the animal experimental facility (Allen Laboratory) at the University of Delaware to be further processed. The field samples were obtained by Dr. Luka Jwander (National Veterinary Research Institute, Vom) and archived under ambient storage conditions, and documented (Table 1).

### 2.4. Importation and DNA Extraction of Samples

Samples obtained on FTA cards were used as the medium for the collection and transportation of the biological specimens was in accordance with importation regulations [78]. These samples were imported under USDA APHIS Importation permit # 134248 to co-author Dr. Brian Ladman, University of Delaware.

DNA isolation was carried out within the Allen Laboratory at the University of Delaware at Biosafety Level 2. For each sample, a sterile hole-punch was used to punch three 2 mm diameter discs from the FTA card into a 1.5 mL sterile micro-centrifuge tube containing 500 µL Whatman FTA^®^ Purification Reagent (Thermo-Fisher, Waltham, MA, USA). The tubes were vortexed and allowed to elute nucleic acids at RT for 15 min. Following this, the tubes were spun for 15 s at 15,000 rpm, and the liquid was transferred to a fresh 1.5 mL tube. To inactivate any potential RNA viruses, 50 µL of 1× TE, RNAse A (25 mg/mL) was added, and the samples were further incubated at RT for 30 min. DNA was then extracted using the PK solution (10 mM Tris-HCl, pH 8.0, 10 mM EDTA, pH 8.0, 100 mM sodium chloride, and 2% SDS (*w*/*v*) + 4 mg/mL Proteinase K), vortexed, and incubated at 56 °C overnight. The DNA was then purified by phenol-chloroform extraction followed by isopropanol precipitation, according to standard methods [79]. The DNA precipitate was washed with 70% ethanol, air dried, and resuspended in 100 µL 1× TE buffer, pH 7.5.

### 2.5. Amplification of meq Genes

We designed two specific PCR primers targeting the translation start and stop codon to amplify the MDV *meq* open reading frame (ORF). The forward primer contains an *Nhe* I site, a MYC epitope tag, and the first 20 nt of the *meq* ORF, and the reverse primer with a *Hind* III site and the last 19 nt of *meq* (Table 2). The PCR mixture contained 50–100 ng of sample DNA, primers (10 μM each), 2× Platinum SuperFi II DNA Polymerase Master Mix (Invitrogen), and nuclease-free water. The amplification reaction was conducted under the following conditions: 95 °C for 5 min followed by 35 cycles at 95 °C for 1 min, 60 °C for 1 min, and 72 °C for 1.5 min, with a final extension of 72 °C for 5 min in a thermal cycler. The PCR products were resolved by agarose gel electrophoresis (0.8% agarose Tris-borate/EDTA gel), detected by ethidium bromide staining (TBE buffer containing 0.5 μg/mL ethidium bromide), and visualized under an ultraviolet light transilluminator.

### 2.6. Cloning of meq and Chicken Nfil3 Genes and Construction of Expression Plasmids

The 1077 bp amplicons containing the *c-myc* epitope tag fused to the *meq* ORF were excised following agarose gel electrophoresis and extracted using the QIAquick PCR Purification Kit (Qiagen, Valencia, CA, USA). The *meq* gene was topo-cloned into the pCR2.1-TOPO vector using the TOPO TA cloning kit (Invitrogen, Thermo Fisher Scientific, Waltham, MA) according to the manufacturer’s instructions. Prior to cloning, PCR products were incubated with Taq at 72 °C for 20 min to add 3′ A-overhangs to increase cloning efficiency. Eight independent clones were screened via *Eco*RI digestion of plasmid DNA after plasmid purification using standard methods (phenol-chloroform extraction and isopropanol precipitation). Positive clones were propagated on a larger scale and purified using a Qiagen Plasmid Purification Kit (Qiagen, Valencia, CA, USA). At least two independent clones per sample were sequenced for single nucleotide variations. The *c-myc-*tagged chicken *Nfil3* ORF was amplified from cDNA templates isolated from MDV-infected chicken specimens and cloned into the pCR2.1-TOPO vector as above.

The *myc*-tagged *meq* cassette was ligated into the pBKCMV vector to construct epitope-tagged expression vectors. For the Meq and NFIL3 C-terminal fusion proteins, fluorescent protein fusion constructs were generated by PCR mutagenesis to remove the stop codon in the coding sequence. Following PCR amplification, the *meq* and *Nfil3* fusion cassette were TA ligated into the pCR2.1 TOPO vector for sequence validation and subsequent sub-cloning. The resulting fusion gene cassette allowed for the insertion of the genes upstream of the fluorescent protein cassette in the pECFP-N1 and pEYFP-N1 vectors.

### 2.7. MDV Genome Copy Number Analysis

The genomic DNA samples extracted from FTA cards were analyzed by real-time quantitative PCR for the detection and quantification of MDV viral genomes per 10,000 cells. DNA stocks were quantified using a nanodrop spectrophotometer and diluted to 50 ng/mL prior to reaction setup. Primer sets were used to either target the MDV040 ORF that codes for the glycoprotein B (gB) of MDV-1 or the chicken ovotransferrin (ova) gene of the host cellular genome, as previously described [80,81]. Quantitative PCR was performed using a Bio-Rad MyiQ2 Two Color Real-Time PCR Detection System (Bio-rad Laboratories, Hercules, CA, USA). The PCR mixture of 20 mL contained 10 mL iTaq Universal SYBR Green Supermix (Bio-Rad, Hercules, CA, USA), 250 nM of each primer, and 50 ng of DNA. The thermal cycling conditions consisted of an initial denaturation at 95 °C for 3 min, followed by 40 cycles of 95 °C for 10 s and 55 °C for 30 s. Using the method to correct for diploid ovo copies in the chicken genome, the ovo gene served as an internal normalization standard to determine cell number [81]. Absolute copies of gB were normalized to chicken ovo to obtain number of viral genomes per 10,000 cells.

### 2.8. Genome Sequencing

MDV-positive samples were processed for next-generation sequencing (NGS) by fragmentation of 5.0 μg of total DNA. Sequence libraries were subsequently prepared using the NEBNext Ultra II Library Prep Kit (New England Biolabs, Ipswich, MA, USA). To specifically enrich viral sequences from infected chicken DNA extracts, a tiling array was utilized as previously described [82]. This array, comprising 6597 biotinylated RNA 80-mers, was tailored based on the RB1B strain sequence (GenBank EF523390.1). The final design was ordered as myBaits Custom Kit (Arbor Biosciences, Ann Arbor, MI, USA). Enrichment was carried out following the manufacturer’s protocol, with the modification of extending the hybridization period to 20 h at 65 °C. In the final amplification step of the enrichment protocol, 12 PCR cycles were conducted using Q5 High-Fidelity DNA Polymerase (New England Biolabs). Enriched libraries were sequenced at the Institut für Virologie, Freie Universität Berlin (Berlin, Germany) on an Illumina MiSeq platform. The genomes were assembled *de novo* using SeqMan NGen (DNASTAR, Madison, WI, USA) and were performed using a reference-guided approach. The preprocessed and quality-trimmed Illumina reads were mapped against the MDV strain RB1B reference sequence (EF523390.1). The sequencing results are summarized in Table 3. Consensus genomes were generated for phylogenetic analysis.

### 2.9. Sanger DNA Sequencing

To validate the candidate variants identified by NGS, each cloned *meq* gene was sequenced on a Sanger dideoxy sequencing platform, the ABI 3500 Genetic Analyzer (Applied Biosystems Inc., Foster City, CA, USA). DNA sequencing was performed at the University of Delaware (University of Delaware DNA Sequencing & Genotyping Center, Delaware Biotechnology Institute, Newark, DE, USA). Bidirectional Sanger sequencing with vector-based primers (M13F and M13R) in conjunction with internal *meq*-specific primers as previously described [34] was used to identify non-synonymous point mutations in the polymorphic regions of the *meq* ORF. The sequences obtained were assembled using the SeqMan Pro program in Lasergene (DNASTAR, Madison, WI, USA) with RB1B *meq* (EF523390.1) as a reference. A consensus sequence with 3-fold coverage at each base pair was generated, and nucleotide variations were manually called.

### 2.10. Sequence Analysis (MSA, Pairwise, aa and nt)

The *meq* nucleotide and protein accessions used in this study were retrieved from GenBank and given in Table 4. Sequence analysis and alignments were performed using the bioinformatics software MeqAlign Pro (DNASTAR, Madison, WI, USA). Local alignment searches were initially made with NCBI BLAST to infer sequence homology. Multiple sequence alignments of nucleotide and protein sequences were generated using MAFFT [83] and Clustal Omega [84], respectively. Distance matrices were generated using Uncorrected Pairwise Distance with Global gap removal metrics and presented as %Identity via the conversion formula stated within brackets (%ID = 100 × (1 − distance)).

### 2.11. Phylogenomic Analysis

To assess the evolutionary relationship of the Meq isoform encoded by Nigerian field strains, a protein sequence dataset composed of the Nigerian Meq isoforms was compared to 26 representative Meq isoforms. The Meq isoforms included in this analysis are from prototype MDVs and parental viruses isolated from the US, Europe, and Asia. The dataset used for the phylogenetic analysis of the Meq isoforms is given in Table 4. For reconstructing MDV-1 evolutionary lineages, a phylogenetic analysis was performed on the five new partial genomes from Nigeria compared to 31 reference MDV-1 partial and complete genomes of different pathotypes and geographical regions of isolation (Table 5). Phylogenetic analysis was performed on these protein and nucleotide datasets with the parameters/ models of evolution stated as follows: (a) the phylogenies were estimated with maximum likelihood optimality criterion which was executed by RAxML v8.2.12 [102], (b) the maximum likelihood phylogenetic trees were construction by incorporating rate heterogeneity using the gamma (Γ) distribution model (GAMMA+P-Invar) along with the general time reversible (GTR) substitution matrix, and (c) support values for phylogenetic relationships were obtained by simultaneously conducting a rapid bootstrap analysis and 1000 bootstrap replicates. MegAlign Pro (DNASTAR, Madison, WI, USA) was used to conduct maximum likelihood phylogenetic analysis and draw trees.

### 2.12. Cell Culture and Transfections

HD11 and HTC cells, both immortalized chicken macrophage cell line [103], were used in this study for transfection experiments. Cell cultures were maintained in high glucose Dulbecco’s modified Eagle’s medium (DMEM) supplemented with 10% FBS (R&D Systems, Inc., Minneapolis, MN, USA), 0.5 μg/mL amphotericin B, and 1× PSN (Gibco, Thermo Fisher Scientific, Waltham, MA, USA) at 37 °C with 5% CO_2_. For colocalization analysis, HD11s were seeded onto 12-well dishes at a plating density of 2 × 10^5^ cells per well and were allowed to reach 60–70% confluency at the time of transfection. Prior to transfection, growth media was replaced with DMEM without antibiotics or antimycotics. Expression constructs were transfected using 1 μg of Lipofectamine 2000 (Invitrogen), and DNA-liposome complexes were prepared in serum- and antibiotic-free DMEM according to the manufacturer’s suggestions. Transient transfections were allowed to fill in overnight before immunofluorescence analysis (IFA).

### 2.13. Antibodies

The rabbit anti-Meq polyclonal serum used was pooled anti-Meq antisera from rabbits immunized with *E. coli*-expressed Meq amino terminus aa 1-106 and was generously provided by Dr. Hans Cheng (USDA-ADOL) and pre-adsorbed against ETOH-fixed chicken cell lines HD11, HTC, CU91, and DF1 cells. Goat anti-rabbit conjugated with Alexa 568 or Alexa 488 and goat anti-mouse Alexa 568 were used as secondary antibodies (Molecular Probes, Thermo Fisher Scientific, Waltham, MA, USA). Antibodies were prepared in antibody diluent (1× PBS, pH 7.4, 3% goat serum, 1% BSA, 0.1% saponin, 0.1% NaN_3_) at a final dilution of 1:100 for primary antibodies and 1:200 for secondary antibodies.

### 2.14. Immunofluorescence Analysis (IFA)

For the localization study, HD11s were transiently transfected with 200 ng of the LEC-LG *meq* expression vectors (pBKCMV-MYC-Meq or pECFP-N1-MYC-Meq) per well in a 12-well plate. After 24 h, transfected cells were fixed with 1% paraformaldehyde in PBS for 30 min, followed by three washes with PBS. Cells were blocked for 1 h in blocking buffer (1× PBS, pH 7.4, 3% goat serum, 1% BSA, 0.1% saponin, 0.1% NaN_3_) prior to staining with the polyclonal antibody to Meq (rabbit serum) or the mouse anti-MYC epitope tag antibody (9E10 hybridoma supernatant) for 2 h at RT. Cells were washed three times with wash buffer (1× PBS, pH 7.4, 1% BSA, 0.1% NaN_3_), and incubated for 1 h with the appropriate secondary antibody (goat anti-mouse IgG, whole molecule Alexa 555 or 568, as noted; or goat anti-rabbit IgG, whole molecule Alexa 488 or 568, as noted). Finally, cells were washed three times, as described above and counterstained with DAPI imaging buffer (1× PBS, pH 7.4, 10% glycerol, 6 nM DAPI, 0.1% NaN_3_). Image acquisition was performed with a Nikon Eclipse TE2000-U inverted epifluorescent microscope with a Plan Fluor 20× objective, Nikon Digital Sight DS-QiMc camera, and Nikon NIS Elements imaging software (v5.02).

### 2.15. Statistical Analysis

Statistical analyses were performed using GraphPad Prism v5.01 (GraphPad Software, Inc., Boston, MA, USA). The growth curves were analyzed using a one-way ANOVA, with Dunnett’s multiple comparisons test compared to the vector cell line control; *p*-values < 0.05 was considered statistically significant.

### 2.16. Sequence Accession Numbers

The nucleotide sequences identified in the present study were deposited into the GenBank database. The designated GenBank accession numbers are indicated in Table 6.

### 2.17. Glycoprotein L (gL) Mutation Assay (PCR-RFLP)

The gL locus was amplified from each of the DNA samples with I7 High-Fidelity DNA polymerase 2× master mix (Intact Genomics, St. Louis, MO, USA), the gL specific pathotyping primers (Table 2), nuclease-free water, and 50 ng of sample DNA in a total reaction volume of 50 μL. The PCR was carried out under the following cycling conditions: 94 °C for 5 min followed by 35 cycles at 94 °C for 1 min, 55 °C for 1 min, and 72 °C for 1 min, a final extension of 72 °C for 5 min in a thermal cycler. The PCR products were subsequently purified by ethanol precipitation and resuspended in 10 μL TE buffer. Restriction fragment length polymorphism (RFLP) was conducted to detect the presence or absence of the 12 nt deletion in the gL gene as previously described [104].

Briefly, the 759/771 bp amplicon containing the 12 nt insertion or deletion (indel) was digested with the *Dde*I restriction endonuclease (New England Biolabs Inc., Beverly, MA, USA) to cleave once or twice at the recognition site that resides in the region of genetic variation. The digested amplicons were resolved on a 1% agarose gel alongside positive and negative controls prepared from TK1a and RB1B viral DNA, respectively. The 12 nt indel was discerned qualitatively by the resultant fragments.

## 3. Results

### 3.1. Gross Pathology

Chickens from one backyard flock and 11 flocks on five commercial broiler and six layer farms were described as having acute MD clinical signs, lesions, and mortality. The farms were located in the Plateau State of North-Central Nigeria, and the suspected MD outbreaks occurred from 2015 to 2016. Vaccination practices on the affected farms indicate that 100% of broiler flocks (n = 6) were unvaccinated against MD, while 100% of layer flocks (n = 6) were vaccinated with commercial MD vaccines. All vaccinated layer flocks were administered CVI988 on the first day-of-age (hatch) before departure from the hatchery. Among the layer flocks, 50% (n = 3) were revaccinated with the HVT vaccine administered at 21 days-of-age by veterinarians on site.

The chickens from flocks suspected of MDV infection presented common clinical signs indicative of acute MD (Table 1). Cases were most severe in the unvaccinated broiler flocks, with veterinarian records indicating stunted growth, emaciation, ruffled feathers, paled mucous membrane, dark mucous feces with enteritis, and prominent keel bone (Table 1). Post-mortem reports indicated overt splenomegaly and hepatomegaly with lymphomatous lesions and gross visceral lymphoma on the heart, liver, and spleen during necropsy of broilers (Table 1). Notably, the unvaccinated flock of backyard broilers (MH1) experienced high levels of mortality, with chickens found dead suddenly overnight (Table 1). The common gross pathological alterations observed during necropsy of vaccinated layer flocks were hepatomegaly with lymphoma and accumulation of sanguineous fluid within the adnominal cavity (Table 1). Other gross tumors of the liver, spleen, lung, heart, ovaries, and kidneys were observed, and solid tumors, along with the surrounding tissue, were collected (Table 1).

### 3.2. Molecular Analysis

To determine the presence of MDV in the tissue and tumor samples obtained, the isolated DNA was examined by endpoint PCR (Meq, gL) and quantitative PCR (gB) (Appendix A). From these DNA samples, we obtained 30/40 (75%) Meq amplicons that were subsequently cloned for sequencing. For subsequent attempts at whole genome sequencing, we performed quantitative PCR (qPCR) based in the gB amplicon to identify samples with the highest genome copy number. Of these, 22/40 (55%) of the samples had detectable genome copy numbers. For the gL mutation assay, we selected DNA samples that were positive for Meq amplicons and were representative of the different farms. Of these, 13/19 (68%) were positive for gL amplification. The total number (percent) of DNA samples for which Meq and/or the gB qPCR reactions were positive was 34/40 (85%).

The differences in the ability to amplify MDV sequences from the FTA-derived DNA samples may be due to the fact that the *meq* gene (RLORF7) is present in two copies per genome and the fact that for qPCR, the amplicon is much smaller and the amplification cycle number increased for this analysis.

### 3.3. Virus Reisolation Attempts

To characterize the pathogenicity of the Nigerian MDV strain, we attempted to isolate infectious virus and whole virus genomes for reconstitution from specimens collected from chickens showing signs of clinical disease. Given that the infected samples were obtained as FTA card specimens, we were unable to successfully isolate or reconstitute the virus by traditional virological methods due to pathogen inactivation and DNA shearing from FTA card matrices. Following this attempt, subsequent efforts to reconstitute the virus by generating an infectious recombinant BAC clone were also unsuccessful.

### 3.4. Whole Genome Sequencing and Draft Genome Assembly

Upon diagnosing suspected MD incidence and with respect to genome copy numbers of infected clinical samples, we used a previously established targeted MDV sequence enrichment [82] to sequence the viral genomes from five flocks experiencing cases of MD, including BRM, CMB, MH2, NGH, and RT. Flocks with severe clinical signs and vaccination history from each vaccination protocol used were represented in this sample set. Partial genomes were assembled as scaffolds ranging in length from 177,635–186,673 bp in length using the RB1B reference genome to guide the assembly. We found the Nigerian strain genomes to resemble other alphaherpesviruses in size and organization into the characteristic class E genome architecture, containing six genomic regions (TRL-UL-IRL-IRS-US-TRS). The details of the five isolates and summary of assembled reads for each genome are given in Table 3.

### 3.5. MDV-1 Phylogenetic Analysis

To determine the phylogeographic relationship and evolutionary lineage of the emergence of new Nigerian field isolates, we built phylogenetic trees using the maximum likelihood approach. MDV genomes are divergent in that the direction of evolution is presumably driven by constraints related to the co-divergence of virus and host lineages, including the differences in international vaccination practices, poultry breeding and genetics, and management practices. According to previous phylogenetic and comparative genomic analysis, the reconstruction of MDV phylogeny based on complete or partial genomes suggests that North American and Eurasian stains emerged from independent evolutionary paths [63,98,105].

We performed sequence alignments on the genomes of 36 international MDV isolates collected between 1964 and 2016 from NA, Europe, Asia, and West Africa. The partial genomes of the Nigerian field isolates present a limitation within our phylogenetic analysis. However, MDV sub-genomic segments can give insight into evolutionary dynamics, where the phylogenetic trees based on the UL subregion support the construction of phylogeny based on complete genomes [63,105]. To account for inaccurate inferences of phylogeny, we also constructed a phylogenetic tree based on sequences of the UL sub-region.

The construction of unrooted phylogenetic trees reveals the monophyletic grouping of strains into NA and Eurasian clades (Figure 1). Within this geographical framework, MH2, CMB, BRM, and RT were grouped with the other Eurasian strains into a clade, whereas NGH clusters in the NA clade due to close genetic relations with the RB1B strain. MH2 and CMB genomes have close genetic relatedness, as do BRM and RT genomes; accordingly, these genomes cluster with recently isolated strains from Europe (Hungary, Israel, and Poland) in the European subclade. The latter two genomes cluster with European strains and share a more recent common ancestor with the hv Polish strain, Polen5 (Figure 1). Notably, MH2 and CMB genomes constitute a distinct subclade separate from BRM, RT, and other Eurasian strains, implying the Nigerian field strains could have evolved independently. Because of the paucity of MDV genomes sequenced from Western Africa, we could not determine if the Nigerian strain conforms to a clade associated with Afro-Eurasia.

The phylogenetic tree based on the complete genomes closely resembles the topology of the tree based on the UL subregions, except for MH2, which shares genetic relatedness to the genome from the CVI988 vaccine strain originally isolated from Europe (Netherlands), and together these strains form a separate clade with RB1B and NGH (Figure 2).

### 3.6. PCR-RFLP Analysis of Glycoprotein L (gL)

The 12 nt insertion or deletion (indel) in the gL gene is a virulence feature associated with USA field strains [34], in which vaccine-mediated pressures select for the deletion that ablates four amino acids at a putative MHC-I signal peptide cleavage site [106] that is encoded by some HVT or bivalent vaccine resistant vv+MDVs [104,107]. Based on the MD outbreak in vaccinated Nigerian flocks, the 12 nt gL deletion was examined by PCR-RFLP (Appendix A).

### 3.7. Comparison of Meq Coding Sequences

To complement the MDV phylogeny and sequence heterogeneity at the genome level, we investigated the *meq* gene sequence of the Nigerian MDV field strains. Given the genetic variability within the *meq* locus of MDV field strains, which typically corresponds to parental strain pathotype, we addressed the polymorphisms that would characterize the level of pathogenicity of circulating strains in Nigeria.

Sanger sequencing analysis of the amplified *meq* genes led to the identification of three distinct isoforms of the *meq* gene among the Nigerian field strains. All mutations identified in *meq* were either synonymous or non-synonymous point mutations. The *meq* nucleotide sequence of the five genomes sequenced by NGS was confirmed by Sanger sequencing and evaluated for sequence homology with representative MDVs. To further validate NGS results, we expanded our analysis by sequencing *meq* from one to two tissue samples collected from each flock suffering from MD. The sequences of *meq* of each isolate were deposited in GenBank (Table 6). Among the Nigerian field isolates, the *meq* genes from 19 clinical samples all contain the 1020 bp ORF encoding the 339 aa Meq as the predominate isoform. The detection of indel genetic variations, particularly the previously identified 177–180 bp insertion [34,108] was not represented in the sample set. The 19 *meq* genes included in this analysis show modest genetic heterogeneity, with the sequence identity ranging from 98.92 to 100% and 97.35 to 100% of positions in aligned nucleotide and amino acid sequences, respectively. BLAST searches found that the predominate *meq* gene detected among the Nigerian field isolates is unique, with 10 of the 19 isolates not represented in the GenBank database, as no identical sequences were found at the time of searching.

Pairwise comparisons of the Meq coding sequences of representative MDVs with the Nigerian isolates are given in Table 7. The nucleotide sequences of isolates CMB, MH2, EB1, EB2, LEC, and MH1 ranged in identity from 98.92 to 99.02% with the RB1B *meq.* The deduced amino acid sequences share 97.35% identity with RB1B Meq (Table 8). Sequence identities across the 10 novel Meq proteins vary from 96.76 to 98.82% according to the representative Meq isoforms, with the highest sequence homology between the Nigerian isolates from flocks CMB, MH2, EB1, EB2, LEC, and MH1 with those of highly virulent European MDVs (C12/130 and ATE2539), followed by vv+ MDVs from the USA.

Additionally, the BLAST searches also found homologous sequences to nine of the 19 *meq* genes encoded by the Nigerian field isolates and those of previously identified MDV strains deposited in the GenBank database. Among these Nigerian field strains, the *meq* genes from the isolates NGH and BBL are 100% identical with the RB1B *meq* at both the nucleotide and amino acid levels, Table 7 and Table 8, respectively.

For isolates BRM, RT, WKB, and GDA-FT, the *meq* genes share 99.61% identity with RB1B *meq*, while the sequence identity at the amino acid level is 98.82%. The sequence identities at the nucleotide and amino acid levels are identical (100%) to Polen5 *meq*, a hypervirulent strain isolated from Poland in 2010 [98].

The recent outbreaks of MD in Nigeria and the severity of disease in vaccinated chickens suggest a high virulence strain in circulation. The emergent MDV field strain encoding this novel Meq isoform was identified in 50% (n = 6) of the infected and diseased flocks. As opposed to Meq isoforms of high virulence parental viruses isolated from NA and Europe, the predominate Meq isoform from Nigerian field isolates (CMB, MH2, EB1, EB2, LEC, and MH1) has accumulated non-synonymous point mutation in the region coding for essential domains within both the amino (N)- and carboxyl (C)-terminal proximity of the Meq protein. Except for two transversions, the non-synonymous point mutations to *meq* were G > A transitions at the nucleotide level, resulting in the amino acid substitutions described herein. By comparing the protein sequence of the canonical Nigerian Meq isoform and the RB1B Meq, we identified a total of nine amino acid substitutions. Position of residue substitution based on alignment of Meq proteins with the canonical Meq sequence from Nigerian field isolate LEC-LG are given in Table 9 and are: K77E, D80Y, A88T, Q93R, T139A, P176A, T180A, P217A, and E263D.

In comparison to vv and hv European strains, the LEC-LG Meq is conserved at distinct amino acid positions within the N-terminal region of Meq. Amino acid residues at positions E77 and Y80 of the N-terminal basic region are conserved across vv and hv European strains (C12/130, ATE2539, EU-1, Polen5, MR48, MR36, and ATE). In the LZ motif, residues at position T88 and R93 are conserved with pC12/130-10, pC12/130-15, ATE2539, and ATE strains. In contrast to European strains, the LEC-LG Meq has conserved substitutions with vv+ USA strains (648A, 686, N, 584a, and TK1a) at residues A176 and A217 in the C-terminal, PRR domain. The substitution of proline residues at 176 and 217 for alanine interrupts the proline tetrads (PPPP) at the second position, thereby reducing the number of proline tetrads to three compared to five in RB1B Meq.

### 3.8. Phylogenetic Analysis of MDV Strains

A phylogenetic analysis was performed using the entire Meq isoform to investigate the molecular phylogenetic relationships among representative MDV strains. The virulence-associated mutations in the *meq* gene have been extensively studied [34,37,38]. Thus, we used this to estimate the pathogenic relatedness of the Nigerian parent strain based on the genetic heterogeneity of the oncogene among prototype strains having pathotype classification. To avoid compositional biases, we aligned the deduced amino acid sequences of 27 Meq genes encoded by NA and Eurasian strains. The *meq* tree reconstructs the divergent evolution of MDV strains, which supports the phylogenetic tree based on complete genomes (Figure 1 and Figure 3). Similar to the partial genome sequence, LEC-LG Meq appears to be a descendant of the European lineage and shares a recent common ancestor with vv and hv MDVs from Europe (C12/130, ATE2539, ATE). Notably, these Meq isoforms have diverged from a common ancestor shared by European and Asian strains. Isolates BRM, RT, WKB, and GDA cluster with European (Polen5 and MR48) strains, while NGH and BBL are identical to RB1B Meq and likely share a common ancestor cluster with North American (RB1B, Md5, 584A, N, and 648A) strains (Figure 3). Given the heterologous sampling of Nigerian strains with either close genetic relatedness to Polen5 or other Eurasian strains at the genome level, the clustering of the canonical Meq isoform with European strains raises the possibility that the Nigerian strain may have an evolutionary predecessor with European origin.

The mutations accumulating in both the bZIP-NTD and the PRR-CTD of the canonical Nigerian Meq isoform, in juxtaposition to the sequence diversity among Meq genes, renders it difficult to distinguish virulence determinants by phylogenetic analysis (Figure 3, Figure 4 and Figure 5). Given the functional modularity of these two domains, we partitioned the Meq protein into two moieties constituting the bZIP domain (Figure 4A) and the transactivation domain (Figure 5A). We constructed ML phylogenetic trees with the partial Meq sequence of USA prototype strains and European and Asian strains.

The phylogeographic diversity among the partial Meq trees and the geographical relationships are generally consistent with the topologies of the complete genome, UL subregion, and full-length Meq phylogenetic trees. However, we can gain more granularity in the pathogenetic relationship among strains. The phylogenetic tree based on the bZIP-NTD revealed two major clades (Figure 4B), with close relatedness in the bZIP domain from all high virulence USA strains and were divided into a single clade, while the other isoforms constitute the other clade. The latter is subdivided with the bZIP domains of high-virulence Eurasian strains clustering together and attenuated/low-virulence strains (mMDV and vMDV) forming a separate subclade. The PRR-CTD phylogenetic tree shows that the CTD among all high-virulence USA strains was closely related and clustered in a low confidence clade, likely having evolved differently from all low-virulence strains (Figure 5B). As expected, the canonical Nigerian Meq bZIP-NTD clusters with vv and hv European (C12/130, ATE2539, ATE) strains, whereas the PRR-CTD shares close relatedness to vv+ NA (N, 584A, and TK1a) strains.

### 3.9. The Nigerian Strain (LEC-LG) meq Localizes to the Nucleus and Nucleolus and Co-Localizes with Chicken NFIL3

In comparison to RB1B Meq, our sequences analysis reveals substitutions in basic region 2 (BR2) at positions 71, 77, and 80 that are conserved by LEC-LG Meq and Meq isoforms of higher virulence strains from Europe (Table 9). Thus, we asked whether these amino acid differences disrupt the nuclear and subnuclear localization dynamics of Meq (Figure 6). In HD11 cells expressing the MYC-tagged LEC-LG Meq, we found Meq localized to the nucleus and accumulated in the nucleolus in abundance, which resulted in a characteristic bulging morphology (Figure 6A,B). This subnuclear compartmentalization of Meq was similarly observed for the LEC-LG Meq fusion to ECFP (Figure 6C). Our results show that LEC-LG Meq localizes to the nucleus and nucleolus, indicating that these three residues do not affect the characteristic localization signaling properties of BR2.

As a bZIP protein, the LZ region of Meq is notable for mediating homodimerization and heterodimeric interactions with JUN, FOS, CREB, ATF, and, more recently, PAR family members [36,109]. Formation of both Meq homo- and heterodimers are required for oncogenic transformation of T-lymphocytes [110,111,112], whereby the intrinsic stability of Meq-Jun heterodimers specifies DNA-binding at AP-1 sites to upregulate transformation-associated target genes [113]. Considering dimerization is an integral component of the functional and structural properties of Meq, we investigated whether the amino acid differences in the bZIP-NTD among Meq isoforms will retain the ability to re-localize the chicken PAR family member NFIL3. The association of JM, RB1B, N, and LEC-LG Meq-EYFP isoforms with NFIL3-ECFP fusion constructs in the nucleus and nucleolus was examined by colocalization analysis. Perinuclear localization of NFIL3-ECFP was observed when expressed in cells alone (Figure 6D).

In cells co-expressing NFIL3-ECFP and each of the Meq-EYFP isoforms reveal that NFIL3 and Meq colocalize in the nucleoplasm and nucleolus irrespective of substitutions in the bZIP-NTD (JM and ATE), PRR-CTD (N strain), or both (LEC-LG) compared to RB1B Meq (Figure 6E–I). These data indicate that mutations in the bZIP-NTD of LEC-LG Meq do not cause anomalous re-localization of Meq dimerization partners such as NIFIL3. However, based on colocalization alone, we cannot exclude that residues at positions 88 and 93 may alter the stability at the interface of LZ interhelical interactions or promiscuity with novel cellular bZIP partners.

Taken together, the above results show that K77E, D80Y, A88T, and Q93R substitutions in the bZIP-NTD of the Nigerian Meq isoform are inconsequential to nuclear localization and subnuclear re-localization dynamics of Meq either as a homo- or heterodimeric complex.

### 3.10. Screen for Adventitious Agents in Representative DNA Samples

With the isolation of MDV strains of potential increased virulence from a field setting, we tested representative DNA samples from each of the farms for the presence of adventitious agents that may have affected vaccine efficacy (CIAV) or are other oncogenic viruses (ALV-J, REV) (Appendix A). The samples examined are noted in Appendix A, and the results are given in Appendix A. Notably, three of the samples (Chinedu-Mari BLR, CMB; ECWA BLR-1 [Aden], EB1; and ECWA BLR-1 [Bilong], EB2), were positive for both chicken infectious anemia virus (CIAV) and reticuloendotheliosis virus (REV).

## 4. Discussion

In Nigeria, the status of susceptibility to MD entails national concern as MD outbreaks affecting commercial poultry farms are on the rise [44], and with reports indicating incessant yearly increases in the frequency of disease [43,46,47,48]. Historically, the adoption of routine MD immunization programs has never been firmly established in Nigeria [50,114]. Farmers seldom use vaccination and revaccination practices due to poor awareness, ad hoc veterinarian consultations, vaccine procurement, and other logistical limitations of live attenuated vaccines [44,53,54,115].

The administration of HVT is more frequently used than any other serotype to protect commercial flocks, followed by CVI988, then SB-1 [53]. This is problematic in high-risk areas for MD outbreaks, such as Plateau State [54], considering HVT does not confer protection against vv or vv+ MDV challenge [4]. Consequently, MD incidence was frequently diagnosed in unvaccinated birds compared to vaccinated birds, but prevalence was high, irrespective of vaccination history [48]. Nevertheless, the increased usage of CVI988 reported in the last decade in Nigeria [115] and the adoption of MD vaccines amongst farmers in Jos, Plateau State, is promising [52].

Concerning the geographic circulation of MDV field strains and the regional dissemination of commercially available vaccines, the introduction of vaccine programs to mitigate MD incidence over the last several decades has been similar in European and Asian poultry enterprises compared to the US poultry industry, due to the earlier integration of the CVI988 vaccine in Europe and Asia. In contrast, the US practiced prolonged usage of HVT and adopted the HVT+SB1 bivalent vaccine before the introduction of CVI988. This dichotomy underlying international vaccination practices contextualizes the MDV phylogeny, where the divergent evolution of NA and Eurasian field isolates reconstructs a monophyletic topology on the basis of geographical relationships [30,63,98,105]. In the present work, the phylogenetic reconstruction of a geographical framework gave the same topology as previously characterized. In this framework, three notable features were revealed by the phylogenetic analysis of the partial genomes of Nigerian field strains. First, NGH clusters in the NA clade and was distantly related to the RB1B genome. Second, BRM and RT are closely related to the Polen5 genome and cluster with other European strains in the Eurasian Clade. Finally, MH2 and CMB genomes share a common evolutionary lineage with the European and Asian strains within the Eurasian clade.

Our findings provide suggestive evidence that the outbreaks in Nigeria were caused by multiple introductions or co-circulation of highly virulent MDV strains due to the heterologous sampling of three genetically distinct viruses. Countries in the Eastern Hemisphere are experiencing increasing MD prevalence [115,116] in addition to changes in pathogenesis as indicated by reports from China and Europe that characterize novel emerging virulent field isolates in HVT- and CVI988-vaccinated commercial flocks [19,26,31,33,61,62,96,117]. Surveillance of these strains is needed, but we predict they, or at least the ancestral strains, are pervasive and capable of spreading into Africa.

Polen5, a hypervirulent Polish strain isolated in 2010, has become an epidemic strain spreading throughout Eurasia and has since been isolated from outbreaks in Italy, Iran, and China [57,58,63,72,73]. The close similarity of BRM and RT with the genome of Polen5 and the absence of the gL mutation suggests the Nigerian strains have European ancestry.

Interestingly, Nigeria’s foundation stock is sourced from Western Europe [118], which raises the possibility that the introduction of the virus into Nigeria is from importing contaminated parent stock (i.e., contaminated shipping materials and packaging). With this proximity to Poland, in addition to the pervasive nature of the Polen5 strain throughout Eurasia and Africa during the sampling period of 2015–2016, our findings inform the potential risk of transboundary transmission of MDV via international dissemination (transport) of poultry products.

The Nigerian strains CMB and MH2 are located in a separate subclade with no other strains, and the genomes have diverged from a common ancestry with Eurasian strains. Therefore, we cannot determine the origin of the Nigerian strains. CMB may have been selected in the context of other infectious agents, as noted below (CIAV, REV). Still, we speculate that the Nigerian strain has evolved in the context of improper vaccination practices over time and has been selected for the mutations in *meq*. Alternatively, the emergence of the Nigerian strains can be a *de novo* recombination event. About the latter, the relatedness of the CVI988 and MH2 within the UL subregion collectively with the co-circulation of the Polen5 strain and RB1B strain, we surmise that the Nigerian field strains are derivatives of NA and European parent strains as a result of recombination events in the UL or US subregions. MH2 was sampled from a CVI988-vaccinated layer flock and may represent a heterogenous stock instead.

Nonetheless, the CVI988 vaccine strain has been shown to recombine with Chinese field strains, in which Polen5 was predicted as a putative minor parent strain [29]. The epidemicity of Polen5, along with recombination events involving other Eurasian strains such as ATE2539 to produce recent Chinese strains, may not only account for the divergence from Chinese strains isolated before the 1990s but also as major parental strains in other recombination events [30,63,105]. These events suggest the contribution of MDV recombination to MDV virulence evolution. Thus, further recombination analysis of Nigerian field strains is warranted.

Another important finding in our study was that canonical Nigerian Meq isoforms shared a common recent ancestor with vv and hv MDV strains from Europe (C12/130, ATE2539, and ATE), implying that the Nigerian field strains have adapted to a higher virulence pathotype. The phylogenetic relationship of parental strains at the genome level (Figure 1 and Figure 2) was congruent with the topologies of the phylogenetic trees reconstructed based on the sequence of the cognate Meq isoform (Figure 3). It has been previously established that the RL region of MDV-1 genomes contains the genes of virulence factors in which genetic diversity among strains can reconstruct phylogeny and relatedness based on pathogenicity [92]. Due to the limitation of partial genome sequence in the present study, we focused our analysis on the polymorphisms in *meq* to accurately estimate the virulence level of Nigerian strains in lieu of a ‘best fit’ pathotyping assay. Further studies with rMDVs encoding the LEC-LG *meq* are needed to confirm the pathogenicity in vivo and to determine whether mutations can provide resistance to CVI988 vaccination.

The Nigerian strain is an apparent example of this vaccine-mediated selection of mutations in *meq*. The composite structure of Meq is composed of a bZIP-NTD and a PRR-CTD, each with discrete functional properties, and both have evolved in different directions. Substitutions at positions 77, 80, 88, 93, 139, 176, 180, 217, and 263, which correspond to distinct residues within these domains, are associated with virulence. Of these substitutions, positions 77 and 80 in the bZIP-NTD and 139 and 176 in the PRR-CTD are under significant positive selection pressures [38]. The NTD of the Nigerian Meq isoform shares a common ancestry with Eurasian strains due to the acquisition of mutations in the proximity of the bZIP region that is conserved in highly pathogenic European strains. By stark contrast, the CTD of the Nigerian Meq reflects a separate evolutionary path by conserving substitutions in the PRR region of vv or vv+ US strains.

The bZIP substitutions may alter the binding promiscuity of dimer partners along with their DNA-binding affinity, while the PRR substitutions likely increase transcriptional activity. We postulate that the pressures that select for mutations in the Nigerian strain are indistinguishable from those in which parental strains from NA or Eurasia evolved. For instance, European strains have been selected in the context of monovalent or bivalent MDV-1 serotype vaccines (HPRS-16 and CVI988). On the other hand, US strains have evolved to overcome monovalent HVT and HVT+SB-1 bivalent commercial vaccines but not protection conferred by CVI988. Although MD vaccine strains are antigenically related, serotype-1, unlike serotype-2 and -3 vaccines, encodes a *meq* homolog. Owing to such divergence in the repeat long region, the pressures imposed by CVI988 vaccination appear to be different than those of HVT and HVT+SB-1 vaccines but mutually elicit pro-inflammatory, type I and II interferon (IFN), and interferon stimulating genes (ISG) as part of the innate immune responses to early infection.

For instance, in HVT and HVT+SB-*1* in ovo-vaccinated chicks, IFN-γ and TLR3 transcripts are upregulated in splenocytes, whereas IFN-β and TLR-21 transcripts are upregulated in the spleen of CVI988 in ovo-vaccinated chicks [119,120]. This raises the possibility that these mutations cumulatively provide an evolutionary advantage to overcome vaccinal immunity, either at the level of innate immunity by subverting effector mechanisms of the host immune system or cellular transformation by leveraging host mechanisms involved in cell cycle regulation, cell proliferation, and apoptosis.

There is uncertainty about the pathotype classification or genotypic composition of circulating strains in Nigeria. However, we suggest that the practice of using a highly effective vaccine at hatch (CVI988), followed by a lower efficacy vaccine (HVT) at 21 days-of-age, selects for increased MDV virulence and selection of highly virulent pathotypes. As a result, attempts to control MD by revaccination schedules with heterologous and homologous vaccines in commercial layers and pullets have been reported in current literature without much success in the protection against Nigerian MDV field strains currently in circulation throughout Plateau State poultry farms [27,51]. Apart from molecular diagnostics, these studies mainly investigate the pathological characteristics of vaccine-resistant strains but inadequately address the molecular or phylogenetic characteristics to determine the virulence level of the causative MDV agent. This is due to a need for genome sequencing of MDV field isolates from Nigeria. Thus, the partial genome sequences drafted in the present study provide the initial molecular and phylogenetic characterization of MDV field strains in West Africa, which will aid in understanding the evolutionary dynamics of circulating strains in Nigeria.

Our finding of adventitious agents (CIAV, REV) at a few of these farms, however, suggest that other viruses present at these locations may have contributed to the lack of MD vaccine efficacy and/or contributed to the tumors observed in these birds. As REV has been associated with a change in MDV replication and pathotype through direct integration in the genome [94,121,122,123], as in the case of the RM1 virus and a Chinese virulent MDV, we did not detect REV sequences in the genomes of CMB for which we had greater than 100X genome coverage (Table 3). Interestingly, only the isolates showing REV co-infection also showed the presence of CIAV (Appendix A); and consequently, these chickens were likely highly immune suppressed, perhaps affecting the MDV field strains isolated from these sites. We do note, however, that the MDV field strain having the unique mutations (LEC) did not contain these agents, again implicating the vaccine strategy in the selection of this strain.

Furthermore, we include the clinical history and main pathological findings of the affected flocks to complement the genetic basis underlying the severity of this case. Standard pathotyping assays are required to assess the virulence level of the Nigerian field strains and their resistance to modern vaccines. However, difficulties in isolating viable viruses from infected tissue specimens mean pathogenicity experiments to characterize the Nigerian field strains are not possible.

There has been no reported increase in MDV pathotype above the vv+MDVs to date. In the present case study, however, we speculate that based on the clinical manifestation of MD with disease and mortality in all 12 flocks irrespective of vaccination status in addition to the mutations in the *meq* oncogene, Nigerian strains exhibit the propensity to increase pathogenic characteristics to a level exceeding the vv+ pathotype classification. Our data suggest this outbreak and the profound pathological features in CVI988-vaccinated and revaccinated flocks were caused by administering vaccines improperly, which has, in turn, led to vaccine failure.

Vaccination history for the affected flocks includes unvaccinated broilers and CVI988 vaccinated layers that were administered the vaccine at the hatchery and, in some cases, were revaccinated with HVT at 21 days-of-age. Administering a second, more protective vaccine than that of the primary vaccine will induce a robust immune response that leads to a good outcome for protection against early field challenge [124,125]. Our data suggest that protection is suboptimal when less effective serotype-2 or -3 are applied after CVI988 immunization.

Nigerian poultry operations commonly use this revaccination practice, and it has been reported once prior in an outbreak in Brescia, Italy, that was associated with excessive mortality in broiler flocks caused by the emergent vvMDV field isolate Crescenti [126]. We reason that the timing and the order in which the vaccine serotype was administered caused immunity failure against highly virulent MDVs and is unsuitable for high-risk areas particularly susceptible to MD. Alternatively, improper vaccination may have exacerbated field conditions and prolonged replication, leading to the vaccine-mediated selection of distinct mutations in the *meq* oncogene and the emergence of new variant strains.

## 5. Conclusions

In summary, we report the sequence of an emergent MDV field strain in Nigeria. Although an increase in MDV virulence surpassing the vv+ pathotype has yet to be determined, within the last two decades, sequence analyses of circulating MDVs continue to show genetic heterogeneity in the Meq locus. In alignment with this trend, our sequence analysis identified a novel Meq isoform from those encoded by ancestral MDV strains that emerged between the 1960s and early 2000s. In the highly pathogenic field strains from this period, the selection for virulence-associated residues in either the NTD or CTD of Meq are mutually exclusive. In contrast, the Nigerian MDV has undergone selection for substitutions in both domains, leaving a mosaic of shared ancestry with NA and Eurasian strains, which may, in fact, have been driven by vaccination.

## Figures and Tables

**Figure 1 viruses-17-00056-f001:**
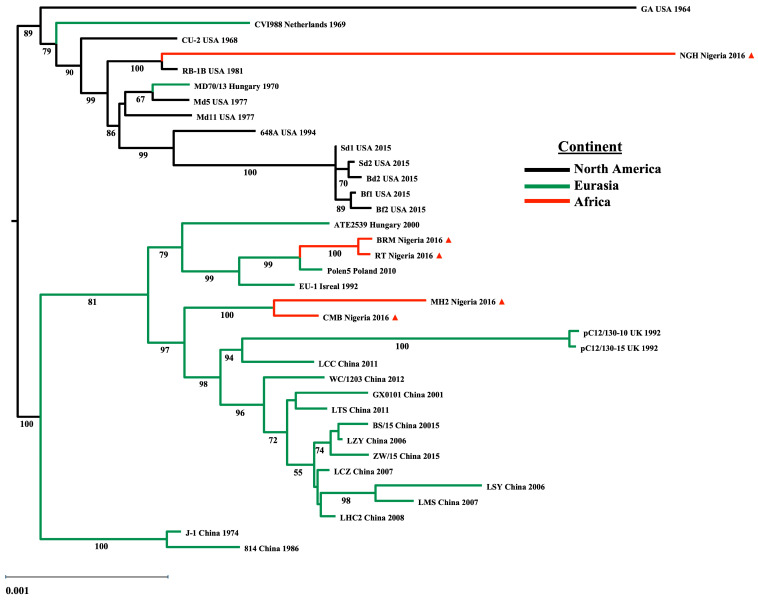
The co-circulating field strains causing the MD outbreak in Nigeria are phylogenetically related to strains from NA and Eurasia. Maximum likelihood estimate tree based on international MDV-1 strains using 36 complete and partial gnomonic sequences, including five sequences from MD positive flocks identified in Plateau State, Nigeria, in 2015–2016 (indicated by the red triangle, ▲) with bootstrap values based on 1000 replications to show the reliability of tree topology. Bootstrap support values were drawn on each node of the tree. Labels include strain, country of isolation, and sampling year. Black = North America, green = Eurasia, and red = Africa. The scale bar represents 0.001 substitutions per codon site.

**Figure 2 viruses-17-00056-f002:**
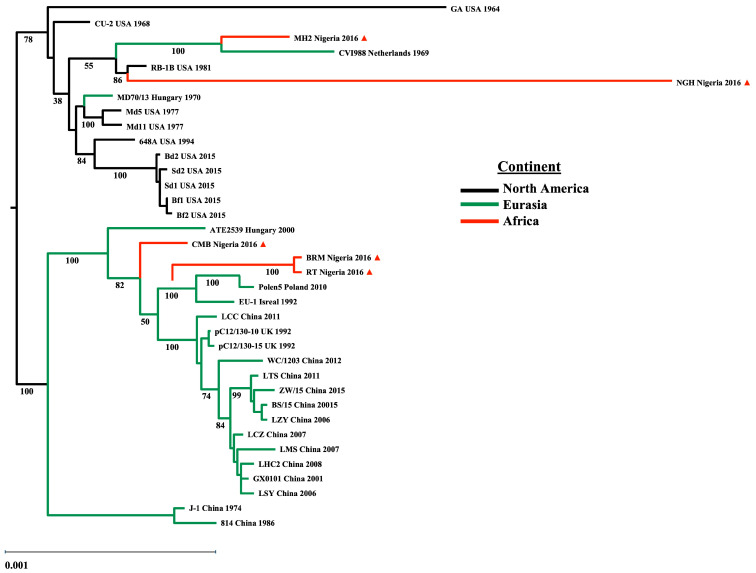
Phylogenetic tree based on the genomic sub-region supports the MDV-1 phylogeny. Maximum likelihood estimate tree of 36 international MDV-1 strains using the unique long (UL) sub-region sequences, including five sequences from MD positive flocks identified in Plateau State, Nigeria, in 2015–2016 (indicated by red triangles ▲). Confidence levels of tree topology were assessed using 1000 bootstrap replications, and support values were drawn on each node of the tree. Labels include strain, country of isolation, and sampling year. Black = North America, green = Eurasia, and red = Africa. The scale bar represents 0.001 substitutions per codon site.

**Figure 3 viruses-17-00056-f003:**
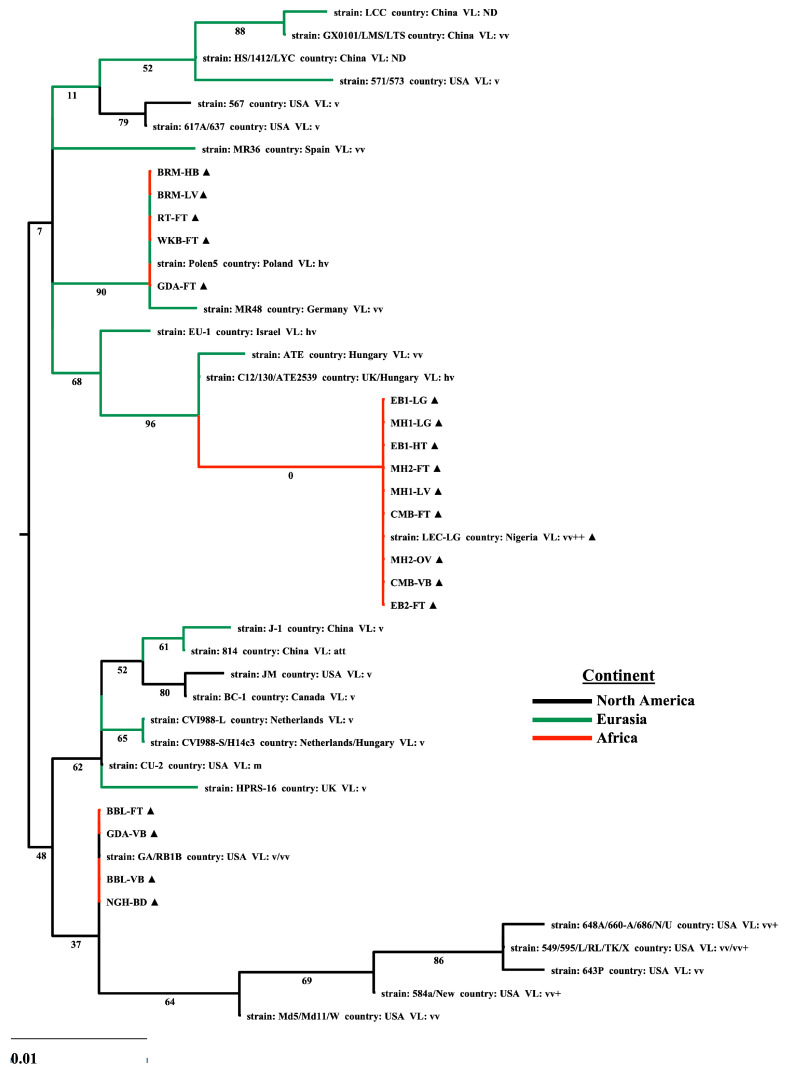
MDV phylogeny and full-length Meq isoforms from strains of North American, Eurasian, and African lineages. Maximum likelihood estimate tree based on the Meq protein coding sequence of Nigerian field strains and 26 representative Meq isoforms encoded by strains from North America, Europe, and Asia. Bootstrap values were based on 1000 replications and were drawn on each node of the tree. Black triangles (▲) indicate the Meq isoforms identified in this study. The parental strain region of isolation for each Meq is represented by branch color: black = North America, green = Eurasia, red = Africa. The scale bar represents 0.01 substitutions per codon site.

**Figure 4 viruses-17-00056-f004:**
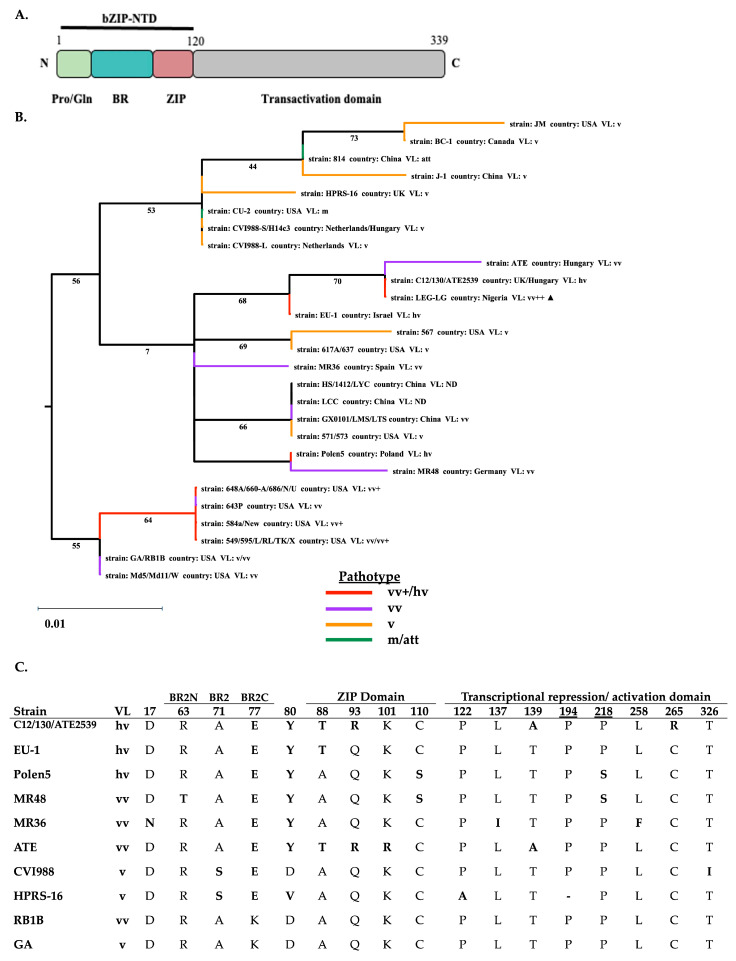
Canonical Nigerian Meq bZIP-NTD has phylogenetic relatedness to highly pathogenic European strains selected in the context of CVI988 vaccination. (**A**) Schematic representation of Meq domain architecture. The bZIP-NTD sequence from 1-120 aa used for the phylogenetic analysis is highlighted. (**B**) Maximum likelihood estimate tree of 27 Meq protein-coding sequences, including the canonical Nigerian Meq (indicated by a black triangle ▲). Bootstrap values were based on 1000 replications and were drawn on each node of the tree. The scale bar represents 0.01 substitutions per codon site. (**C**) Substitution table of European Meq protein sequences was globally aligned with substitutions in reference to the RB1B Meq indicated in boldface. (Pro/Gln: proline/glutamine-rich region, BR: basic region, ZIP: zipper motif).

**Figure 5 viruses-17-00056-f005:**
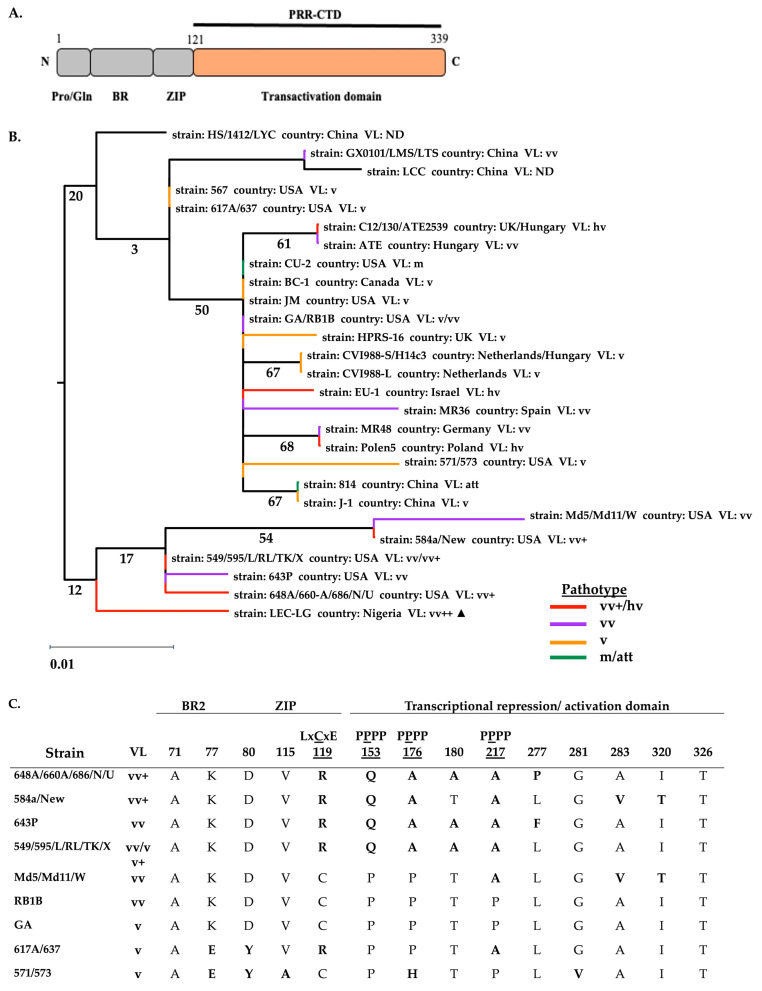
Canonical Nigerian Meq PRR-CTD has phylogenetic relatedness to vv and vv+ USA strains selected in the context of HVT and HVT+SB-1 vaccination. (**A**) Schematic representation of Meq domain architecture. The PRR-CTD sequence from 121–339 aa used for the phylogenetic analysis is highlighted. (**B**) Maximum likelihood estimate tree of 27 Meq protein-coding sequences, including the canonical Nigerian Meq (indicated by a black triangle ▲). Bootstrap values were based on 1000 replications and were drawn on each node of the tree. The scale bar represents 0.01 substitutions per codon site. (**C**) Substitution table of USA Meq protein sequences was globally aligned with substitutions in reference to the RB1B Meq indicated in boldface.

**Figure 6 viruses-17-00056-f006:**
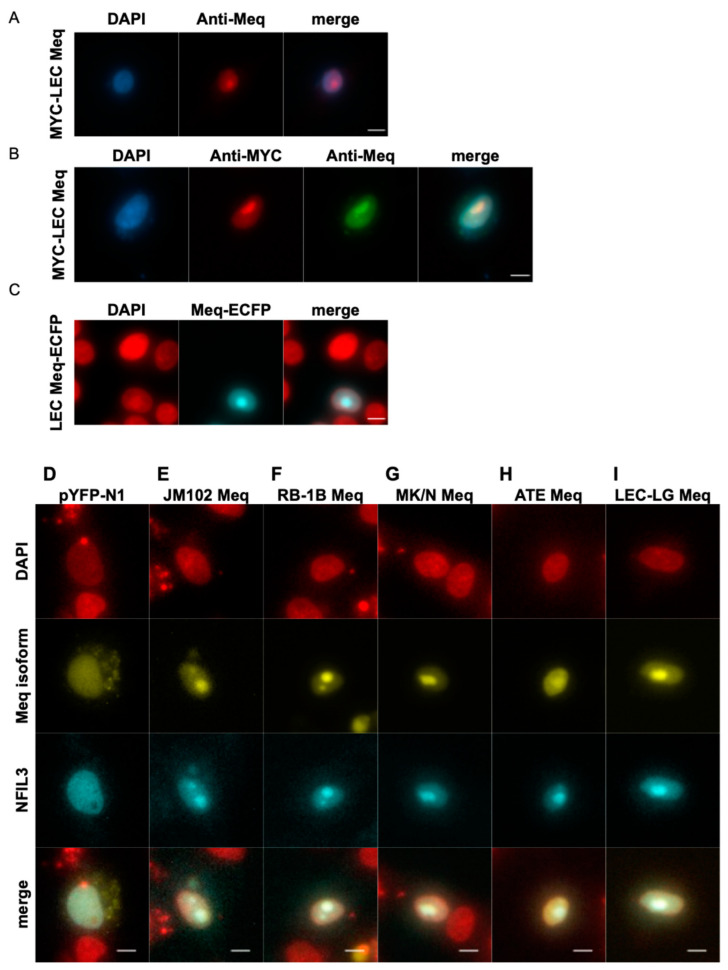
Localization and re-localization dynamics of the Nigerian Meq isoform. Chicken macrophage HD11 cells were transfected with the (**A**,**B**) MYC-tagged LEC-LG or (**C**) ECFP fusion LEC-LG Meq expression constructs and nuclei stained with DAPI. Cells were counter-stained with antibodies to (**A**,**B**) Meq and (**B**) the c-MYC epitope tag. HD11 cells expressing the (**D**–**I**) ECFP fusion chicken NFIL3 or (**E**–**I**) co-expressing EYFP fusion LEC-LG Meq fusion expression constructs. Total magnification = ×300. Scale bars represent 10 μm.

**Table 1 viruses-17-00056-t001:** Description of Source of Nigerian Chicken Samples.

Farm Site ^a^	Flock ID ^b^	Host of Origin	Tissue Samples ^c^	Vaccination History ^d^				Clinical Signs and Gross Pathological Findings
				First Vaccine		Second Vaccine		
				Strain	Age (days)	Strain	Age (days)	
Chinedu-Mari BLR	CMB	Broilers *	LV, SP, VB ^‡^, FT ^‡^	-	-	-	-	stunted growth, emaciation, pale mucous membrane, prominent keel bone, dark mucous feces with enteritis
Brown Mari	BRM	Layers *	LV ^‡^, HB ^‡^	CVI988	1	HVT	21	NA ^e^
ECWA BLR-1 (Aden)	EB1	Broilers *	SP, LV, LG ^‡^, HT ^‡^	-	-	-	-	stunted growth, ruffled feathers, emaciation, pale mucous membrane, tumor on heart, liver, and spleen
ECWA BLR-1 (Bilong)	EB2	Broilers *	FT ^‡^, VB, HT	-	-	-	-	stunted growth, emaciation, ruffled feathers, pale mucous membrane, tumor lesions on liver
NGH (Rantya)	NGH	Layers *	FT, BD ^‡^	CVI988	1	HVT	21	NA
Lay ECWA (Chijoke)	LEC	Layers *	LV, HT, SP, LG ^‡^	CVI988	1	HVT	21	hepatomegaly, serious sanguineous fluid in abdominal cavity
MH1 Mangu	MH1	Broilers ^†^	HT, LV ^‡^, SP, LG ^‡^	-	-	-	-	found dead overnight
MH2 Haija	MH2	Layers *	OV ^‡^, KD, FT ^‡^	CVI988	1	-	-	NA
5 wk BLR (Helen)	WKB	Broilers *	BD, FT ^‡^, LV, SP	-	-	-	-	on postmortem, liver and spleen were enlarged
RT (Morris Rantya)	RT	Layers *	FT ^‡^, BD, LV, HT	CVI988	1	-	-	NA
Gangang Dashe (G. Dashe)	GDA	Layers *	FT ^‡^, VB ^‡^	CVI988	1	-	-	NA
BAM BLR (Nambam)	BBL	Broilers *	VB ^‡^, FT ^‡^, SP, LV	-	-	-	-	No clinical or gross lesions

^a^ The 12 poultry farms located in Plateau State, Nigeria experiencing MD incidence and excessive losses to MD. ^b^ Farm site abbreviations for composing unique sample identifier used in this study. ^c^ Infected tissue samples collected and tumor dissemination: *LV* liver, *SP* spleen, *FT* feather pulp, *LG* Lung, *HT* Heart, *OV* ovary, *KD* kidney, *VB* venous blood, *HB* heart blood, *BD* blood. ^d^ The vaccination schedule and vaccines administered at the sites, unvaccinated flocks were not vaccinated against MD, revaccinated flocks were immunized at hatch with CVI988/Rispens followed by HVT FC-126 strain vaccine at 21 days-of-age, vaccinated flocks were administered CVI988/Rispens the day of hatch at the hatchery. ^e^ NA, not available or not provided. * Indicates commercially improved lines; ^†^ Indicates backyard indigenous broiler breeds; ^‡^ Indicates confirmed positive for MDV-1 by molecular detection and diagnoses as well as *meq* sequence obtained.

**Table 2 viruses-17-00056-t002:** Primers used for qPCR and expression vector construction.

Gene Target	Target Sequence Position	Primer Sequence	Amplicon Size (bp)	Reference Sequence
MDV-1 gB (GaHV-2)	63,997–64,020	F 5′-ATCATTCAGACGACGACATGGACG-3′	91	EF523390.1
	63,930–63,953	R 5′-TGATCTCTTCAATGGAAACGGGCG-3′		
Ch-ovotransferrin	4360–4378	F 5′-CACTGCCACTGGGCTCTGT-3′	71	NC_052540.1/CM028490.1
	4410–4430	R 5′-GCAATGGCAATAAACCTCCAA-3′		
*meq*	136,518–136,537	F 5′-GGCTAGCATGGAACAAAAACTCATCT CAGAAGAGGATCTGATGTCTCAGGAGCCAGAGCC-3′	1077	EF523390.1
	137,528–137,546	R 5′-CCCAAGCTTGCGGATCATCAGGGTCTC-3′		
*meq* fusion		R 5′-GCTCTAGAGCAAGCTTGCGGAATCCTCCGGGTCTCC-3′	1085	
MDV-1 gL	19,731–19,754	F 5′-CTTCACGTATTGTTATGGGTATTT-3′	771/759	EF523390.1
	20,478–20,501	R 5′-TTATCCGATGAAGGTGATGTATCA-3′		
*meq* locus	136,175–136,199	F 5′-AAAGACGATAGTCATGCATGACGTG-3′	1701/1878	EF523390.1
	137,848–137,875	R 5′-TGTATAGAACGAGAATTTGCCATTTAAG-3′		
Chicken NFIL3		F 5′-*GCTAGC*ATGCAGCTGAGAAAAATGCAGAC-3′	1377	AAK72227.1
		R 5′-*GGTACC*TCA*GGATCC*AAGCTTCAGATCCTCTTCTG AGATGAGTTTTTGTTC-ACCAGAGTCTGATGCAGAA-3′		

**Table 3 viruses-17-00056-t003:** Genome assembly and read summary of MDV-1 isolates.

MDV-1 Isolates	Farm Site	Total Reads Number	Assembled Reads Number	Scaffold Length	Coverage Depth
BRM	Brown Mari	493,070	384,591	177,635	163.74
CMB	Chinedu-Mari BLR	374,276	233,511	186,673	108.58
MH2	MH2 Haija	78,666	57,259	178,145	27.66
NGH	NGH (Rantya)	9388	2458	177,960	1.95
RT	RT (Morris Rantya)	1,096,414	834,621	186,598	392.62

**Table 4 viruses-17-00056-t004:** Meq isoforms and MDV strains.

Protein Accession	Nucleotide Accession	Strain	Country	Virulence ^a^ (VL)	Year Isolated	Reference
AAB48631.1 AAP06938.1	M89471.1 AY243332.1	GA RB1B	USA	v vv	1964 1981	[22,34,85,86,87]
AAP06937.1	AY243331.1	JM ^b^	USA	v	1962	[34,88]
AAP06941.1 -	AY243335.1 -	CVI988-S H14c3	Netherlands Hungary	v	1969 1982	[13,34]
AAP06943.1	AY243337.1	CVI988-L ^b^	Netherlands	v	1969	[13,34]
AAR13319.1	AY362707.1	BC-1 ^b^	Canada	v	1955	[34,89,90]
AAR13320.1	AY362708.1	CU-2 ^b^	USA	m	1968	[34,91]
AAR13321.1	AY362709.1	567	USA	v	-	[34]
AAR13322.1 AAR13323.1	AY362710.1 AY362711.1	571 573	USA	v	1989 -	[34]
AAR13324.1 AAR13325.1	AY362712.1 AY362713.1	617A 637	USA	v	1993 -	[34]
AAR13326.1 AAR13327.1 AAR13329.1 AAR13332.1 AAR13333.1 AAR13336.1	AY362714.1 AY362715.1 AY362717.1 AY362720.1 AY362721.1 AY362724.1	549 595 L RL TK1a X	USA	vv vv+	1987 1991 - 1993 1993 -	[4,34]
AAR13328.1	AY362716.1	643P	USA	vv	1994	[4,34]
AAR13337.1 AAR13338.1 AAR13339.1 AAR13330.1 AAR13334.1	AY362725.1 AY362726.1 AY362727.1 AY362718.1 AY362722.1	648A 660-A 686 N U	USA	vv+	1994 1995 1999 - -	[4,5,34]
ABG22688.1 AAR13331.1	DQ534532.1 AY362719.1	584a New	USA	vv+	1990 1999	[4,5,92]
AAS78589.1	AY571784.1	ATE	Hungary	vv	2004	
AFX97850.1 AEZ51745.1 ALA98838.1	JX844666.1 JQ314003.1 KP888838.1	GX0101 LMS LTS	China	vv	2001 2007 2012	[33,62,93,94]
ACR02853.1 AUB50976.1	FJ436096.1 MF431493.1	C12/130 ATE2539	UK Hungary	hv	1992 2000	[95,96,97,98]
AQN78222.1 AEM63536.1	KU744561.1 HQ658627.1	HS/1412 LYC	China	ND	2014 2006	[30,32]
AEV55050.1	JF742597.1	814 ^b^	China	att	1986	[18]
ALA98815.1	KP888815.1	LCC	China	ND	2011	[62]
AQN77176.1	KU744555.1	J-1	China	v	1974	[30]
AUB51061.1	MF431494.1	EU-1	Israel	hv	1992	[98]
AUB51231.1	MF431496.1	Polen5	Poland	hv	2010	[98]
YP_001033993.1 AAS01627.1 AAR13335.1	NC_002229.3 AY510475.1 AY362723.1	Md5 Md11 W	USA	vv	1977 1977 1999	[20,34,99,100]
WYC13990.1	OR592064.1	LEC-LG ^c^	Nigeria	ND	2015–2016	
this paper	-	HPRS-16	UK	v	1967	[101]
this paper	-	MR36	Spain	vv	1994–1995	[31,96,97]
this paper	-	MR48	Germany	vv	1994–1995	[31,96,97]

^a^ Abbreviations: ND Not Determined, att attenuated, m mild, v virulent, vv very virulent, vv+ very virulent +, hv hypervirulent. ^b^ Indicates the strains with the 398 amino acid Meq isoform containing the 59 amino acid proline-rich repeat insertion. ^c^ Canonical Nigerian Meq isoform representing CMB-VB, CMB-FT, EB1-LG, EB1-HT, EB2-FT, MH1-LV, MH1-LG, MH2-FT, MH2-OV. Note: Dotted lines separating rows indicate strains having unique Meq coding sequences, those strains within a dotted row have identical Meq coding sequences.

**Table 5 viruses-17-00056-t005:** MDV genome sequences used in this study.

Strain	Accession No.	Year Isolated	Country	Pathotype	PubMed ID
Md11	AY510475	1977	USA	vv	16155725
Md5	AF243438	1977	USA	vv	10933706
RB1B	EF523390	1981	USA	vv	17721813
GA	AF147806	1964	USA	v	2836620
CU-2	EU499381	1968	USA	m	17557133
648A	JQ806361	1994	USA	vv+	22923089
Bd2	KU173119	2015	USA	hv	Unpublished
Bf2	KU173118	2015	USA	hv	Unpublished
Bf1	KU173117	2015	USA	hv	Unpublished
Sd1	KU173116	2015	USA	hv	Unpublished
Sd2	KU173115	2015	USA	hv	Unpublished
J-1	KU744555	1974	China	v	27112385
LCC	KU744556	2011	China	vv+	27112385
LTS	KU744557	2011	China	vv+	27112385
BS/15	MW247181	2015	China	vv+	28368367
LCZ	MW247188	2007	China	v	Unpublished
LHC2	MW247189	2008	China	vv+	Unpublished
LSY	MW247190	2006	China	v	Unpublished
LZY	MW247192	2006	China	v	Unpublished
WC/1203	KU744558	2012	China	vv	Unpublished
ZW/15	MW247196	2015	China	vv+	Unpublished
GX0101	JX844666	2001	China	vv	23166235
814	JF742597	1986	China	m	21984218
LMS	JQ314003	2007	China	vv	22476905
MD70/13	MF431495	1970	Hungary	v	29151863
ATE2539	MF431493	2000	Hungary	vv+	29151863
EU-1	MF431494	1992	Italy	hv	29151863
CVI988	DQ530348	1969	Netherlands	m	17374751
Polen5	MF431496	2010	Poland	hv	29151863
C12/130-10	FJ436096	1992	United Kingdom	hv	21450941
C12/130-15	FJ436097	1992	United Kingdom	hv	21450941
NGH		2016	Nigeria	ND	This study
BRM		2016	Nigeria	ND	This study
RT		2016	Nigeria	ND	This study
MH2		2016	Nigeria	ND	This study
CMB		2016	Nigeria	ND	This study

**Table 6 viruses-17-00056-t006:** Sequences and Accession Numbers of Nigerian MDV Meq Sequences.

Farm Site	Flock ID	Host of Origin	Tissue Sample	Isolate ID	Accession No.
Chinedu-Mari BLR	CMB	Broiler *	Venous blood	CMB-VB	OR592056
Chinedu-Mari BLR	CMB	Broiler *	Feather pulp	CMB-FT	OR592057
Brown Mari	BRM	Layer ^†^	Liver tumor	BRM-LV	OR592058
Brown Mari	BRM	Layer ^‡^	Heart blood	BRM-HB	OR592059
ECWA BLR-1 (Aden)	EB1	Broiler *	Lung	EB1-LG	OR592060
ECWA BLR-1 (Aden)	EB1	Broiler *	Heart tumor	EB1-HT	OR592061
ECWA BLR-1 (Bilong)	EB2	Broiler *	Feather pulp	EB2-FT	OR592062
NGH (Rantya)	NGH	Layer ^‡^	Blood	NGH-BD	OR592063
Lay ECWA (Chijoke)	LEC	Layer ^‡^	Lung	LEC-LG	OR592064
MH1 Mangu	MH1	Broiler *	Liver tumor	MH1-LV	OR592065
MH1 Mangu	MH1	Broiler ^§^	Lung	MH1-LG	OR592066
MH2 Haija	MH2	Layer ^†^	Feather pulp	MH2-FT	OR592067
MH2 Haija	MH2	Layer ^†^	Ovarian tumor	MH2-OV	OR592068
5 wk BLR (Helen)	WKB	Broiler *	Feather pulp	WKB-FT	OR592069
RT (Morris Rantya)	RT	Layer ^†^	Feather pulp	RT-FT	OR592070
Gangang Dashe (G. Dashe)	GDA	Layer ^†^	Feather pulp	GDA-FT	OR592071
Gangang Dashe (G. Dashe)	GDA	Layer ^†^	Venous blood	GDA-VB	OR592072
BAM BLR (Nambam)	BBL	Broiler *	Venous blood	BBL-VB	OR592073
BAM BLR (Nambam)	BBL	Broiler *	Feather pulp	BBL-FT	OR592074

* Unvaccinated commercial broilers; ^†^ CVI988 (1 dph) vaccinated commercial layers; ^‡^ CVI988/HVT (1 dph/ 21 dph) vaccinated commercial layers; ^§^ unvaccinated backyard broiler breeds.

**Table 7 viruses-17-00056-t007:** Nucleotide identity matrix of the *meq* gene of Nigeria field strains and reference strains.

MD Sample ID	%Identity									
	CVI988	C12/130	Polen5	ATE	GA	RB1B	Md5	643P	New	N
CMB-VB	98.92	99.51	99.02	99.41	98.92	99.02	98.92	99.12	98.92	99.12
CMB-FT	98.82	99.41	98.92	99.31	98.82	98.92	98.82	99.02	98.82	99.02
BRM-LV	99.51	99.51	100.00	99.41	99.51	99.61	99.31	98.92	98.92	98.92
BRM-HB	99.51	99.51	100.00	99.41	99.51	99.61	99.31	98.92	98.92	98.92
EB1-LG	98.92	99.51	99.02	99.41	98.92	99.02	98.92	99.12	98.92	99.12
EB1-HT	98.92	99.51	99.02	99.41	98.92	99.02	98.92	99.12	98.92	99.12
EB2-FT	98.92	99.51	99.02	99.41	98.92	99.02	98.92	99.12	98.92	99.12
NGH-BD	99.71	99.51	99.61	99.41	99.90	100.00	99.71	99.31	99.31	99.31
LEC-LG	98.82	99.41	98.92	99.31	98.82	98.92	98.82	99.02	98.82	99.02
MH1-LV	98.92	99.51	99.02	99.41	98.92	99.02	98.92	99.12	98.92	99.12
MH1-LG	98.92	99.51	99.02	99.41	98.92	99.02	98.92	99.12	98.92	99.12
MH2-FT	98.82	99.41	98.92	99.31	98.82	98.92	98.82	99.02	98.82	99.02
MH2-OV	98.92	99.51	99.02	99.41	98.92	99.02	98.92	99.12	98.92	99.12
WKB-FT	99.51	99.51	100.00	99.41	99.51	99.61	99.31	98.92	98.92	98.92
RT-FT	99.51	99.51	100.00	99.41	99.51	99.61	99.31	98.92	98.92	98.92
GDA-FT	99.51	99.51	100.00	99.41	99.51	99.61	99.31	98.92	98.92	98.92
GDA-VB	99.71	99.51	99.61	99.41	99.90	100.00	99.71	99.31	99.31	99.31
BBL-VB	99.71	99.51	99.61	99.41	99.90	100.00	99.71	99.31	99.31	99.31
BBL-FT	99.71	99.51	99.61	99.41	99.90	100.00	99.71	99.31	99.31	99.31

**Table 8 viruses-17-00056-t008:** Amino acid sequence identity matrix of the Nigerian Meq proteins of those of reference strains.

MD Sample ID	%Identity											Number of PPPPs
	C12/130 ^a^	Polen5	MR48	MR36	ATE ^a^	N ^b^	584a ^c^	549 ^d^	643P	Md5 ^e^	RB1B ^f^	
CMB-VB	98.82	97.35	97.05	97.05	98.53	97.35	96.76	97.64	97.35	97.05	97.35	3
CMB-FT	98.82	97.35	97.05	97.05	98.53	97.35	96.76	97.64	97.35	97.05	97.35	3
BRM-LV ^†^	98.53	100.00	99.71	98.53	98.23	97.05	97.05	97.35	97.05	97.94	98.82	4
BRM-HB ^†^	98.53	100.00	99.71	98.53	98.23	97.05	97.05	97.35	97.05	97.94	98.82	4
EB1-LG	98.82	97.35	97.05	97.05	98.53	97.35	96.76	97.64	97.35	97.05	97.35	3
EB1-HT	98.82	97.35	97.05	97.05	98.53	97.35	96.76	97.64	97.35	97.05	97.35	3
EB2-FT	98.82	97.35	97.05	97.05	98.53	97.35	96.76	97.64	97.35	97.05	97.35	3
NGH-BD ^†^	98.53	98.82	98.53	98.53	98.23	98.23	98.23	98.53	98.23	99.12	100.00	5
LEC-LG^†^	98.82	97.35	97.05	97.05	98.53	97.35	96.76	97.64	97.35	97.05	97.35	3
MH1-LV	98.82	97.35	97.05	97.05	98.53	97.35	96.76	97.64	97.35	97.05	97.35	3
MH1-LG	98.82	97.35	97.05	97.05	98.53	97.35	96.76	97.64	97.35	97.05	97.35	3
MH2-FT ^‡^	98.82	97.35	97.05	97.05	98.53	97.35	96.76	97.64	97.35	97.05	97.35	3
MH2-OV ^‡^	98.82	97.35	97.05	97.05	98.53	97.35	96.76	97.64	97.35	97.05	97.35	3
WKB-FT	98.53	100.00	99.71	98.53	98.23	97.05	97.05	97.35	97.05	97.94	98.82	4
RT-FT ^‡^	98.53	100.00	99.71	98.53	98.23	97.05	97.05	97.35	97.05	97.94	98.82	4
GDA-FT ^‡^	98.53	100.00	99.71	98.53	98.23	97.05	97.05	97.35	97.05	97.94	98.82	4
GDA-VB ^‡^	98.53	98.82	98.53	98.53	98.23	98.23	98.23	98.53	98.23	99.12	100.00	5
BBL-VB	98.53	98.82	98.53	98.53	98.23	98.23	98.23	98.53	98.23	99.12	100.00	5
BBL-FT	98.53	98.82	98.53	98.53	98.23	98.23	98.23	98.53	98.23	99.12	100.00	5

Same superscripts denote Meqs with an identical amino acid sequence and with that of the following strains: ^a^ ATE2539, ^b^ 648A, 660-A, 686, and U, ^c^ New, ^d^ 595, L, R, L, TK1a, and X, ^e^ Md11 and W, ^f^ GA; ^†^ Denotes revaccinated flocks; ^‡^ Denotes flocks that were administered CVI988/Rispens the day of hatch at the hatchery. Shading indicates the canonical Nigerian Meq sequences.

**Table 9 viruses-17-00056-t009:** Amino acid substitutions in Meq isoforms of Nigerian isolates and reference strains.

		Basic Region		ZIP							Transcriptional Repression/Activation Domain										
						LxCxE						PPPP	PPPP		PPPP						
**Strain**	**VL**	**77**	**80**	**88**	**93**	**101**	**110**	** 119 **	**122**	**137**	**139**	** 153 **	** 176 **	**180**	** 217 **	**218**	**258**	**263**	**277**	**283**	**320**
**C12/130/** **ATE2539**	hv	**E**	**Y**	**T**	**R**	K	C	C	P	L	A	P	P	T	P	P	L	E	L	A	I
**Polen5 ^a^**	hv	**E**	**Y**	A	Q	K	S	C	P	L	T	P	P	T	P	S	L	E	L	A	I
**MR48**	vv	**E**	**Y**	A	Q	K	S	C	P	L	T	P	P	T	P	S	L	E	L	A	I
**MR36**	vv	**E**	**Y**	A	Q	K	C	C	P	I	T	P	P	T	P	P	F	E	L	A	I
**ATE**	vv	**E**	**Y**	**T**	**R**	R	C	C	P	L	A	P	P	T	P	P	L	E	L	A	I
**LEC-LG ^b^**	vv++?	** E **	** Y **	** T **	** R **	K	C	C	P	L	** A **	P	** A **	** A **	** A **	P	L	** D **	L	A	I
**648A/660-A** **/686/N/U**	vv+	K	D	A	Q	K	C	R	P	L	T	Q	**A**	A	**A**	P	L	E	P	A	I
**584a/New**	vv+	K	D	A	Q	K	C	R	P	L	T	Q	**A**	T	**A**	P	L	E	L	V	T
**549/595/L/** **RL/TK/X**	vv/ vv+	K	D	A	Q	K	C	R	P	L	T	Q	**A**	**A**	**A**	P	L	E	L	A	I
**643P**	vv	K	D	A	Q	K	C	R	P	L	T	Q	**A**	**A**	**A**	P	L	E	F	A	I
**Md5/Md11/W**	vv	K	D	A	Q	K	C	C	P	L	T	P	P	T	**A**	P	L	E	L	V	T
**RB1B ^c^**	vv	K	D	A	Q	K	C	C	P	L	T	P	P	T	P	P	L	E	L	A	I

Boldface amino acids denote substitutions from RB1B Amino acid substitutions Meq; ^a^ Indicates BRM-LV, BRM-HB, WKB-FT, RT-FT, GDA-FT; ^b^ Indicates CMB-VB, CMB-FT, EB1-LG, EB1-HT, EB2-FT, MH1-LV, MH1-LG, MH2-FT, MH2-OV; ^c^ Indicates NGH-BD, GDA-VB, BBL-VB, BBL-FT; Shading is provided to denote mutations common to European strains (above) and US strains (below) the Nigerian strain.

## Data Availability

All raw DNA samples, sequences and images are available upon request.

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
