# Peer review of "The Meq Genes of Nigerian Marek’s Disease Virus (MDV) Field Isolates Contain Mutations Common to Both European and US High Virulence Strains"

_viruses, 2024, doi:10.3390/v17010056_

Round 1
Reviewer 1 Report
Comments and Suggestions for Authors
The authors carried out the first molecular study on Marek’s disease virus strains causing severe pathology in vaccinated and non-vaccinated chickens in West Africa. The study is very thorough, sequencing the meq genes/proteins of all isolates and the partial genome of some of them to examine molecular characteristics associated with virulence and pathogenicity, while also cloning and expressing the meq to investigate the effects of mutations on the nuclear localization of this oncoprotein, and interaction with host transcription factors. The in-depth sequence analyses of the meq gene and phylogenetic trees shed light on the virulence of the Nigerian isolates and their possible origins.
This is an excellent paper, and I enjoyed reading it. In recent years, there have been numerous publications on molecular characterization and phylogenetic analysis (based on meq sequence) of MDVs from various countries, but this is the most thorough paper I have seen, with the Introduction and Discussion comprehensively summarising the key points from these other papers and bringing all the information together. The paper is written very clearly, in a logical order, and is easy to read and understand. The work has been conducted carefully with appropriate controls. The Methods and Results are presented very clearly and in good detail, and the figures and tables are clear and useful. The results showed that the novel meq gene of the Nigerian MDVs shares ancestry with both North American MDV strains and Eurasian MDV strains, and the authors suggest the combination of mutations could indicate that evolution of new MDV strains with unprecedented virulence levels may already be in progress. I found the Discussion fascinating, and I learned a lot. The paper will be of interest to all those with an interest in MDV molecular biology, virulence, ancestry, evolution, and the pros and cons of control of MD by vaccination.
I have no major comments, just some suggestions for corrections of some words and improving clarity. I have listed these below.
(1) Lines 51 – 57: It would be useful to mention the species names Gallid alphaherpesvirus 2 (GaHV2), Gallid alphaherpesvirus 3 (GaHV3), and Meleagrid alphaherpesvirus 1 (in addition to the older nomenclature MDV-1, MDV-2, and HVT) because the abbreviations GaHV2, GaHV3, and MeHV-1 (and also MDV-3) are used in Table 2 but (as far as I can see) have not been defined.
(2) Line 55: SB-1 is an MDV-2 strain (not MDV-1).
(3) Line 60: I suggest changing ‘upon’ to ‘after’.
(4) Line 70: ‘standard’ Meq is 339 aa, but it would be useful to mention the sizes of the shorter and longer isoforms here.
(5) Line 98: Change ‘1016’ to ‘2016’.
(6) Line 134: The abbreviation ‘DOCs’ is (I think) ‘day-old chicks’, but should be defined here.
(7) Line 156: I think the word ‘and’ is missing (‘Importation and DNA…..’).
(8) Table 1: The abbreviation ‘NA’ has not been defined.
(9) Table 1: Change ‘mucus membrane’ to ‘mucous membrane’ (as in line 374).
(10)It could be useful to mention the chicken Nfil3 protein in the Introduction (or Methods) because I had to wait until the Discussion to find out what it is.
(11)Table 2: It is not clear to me when the MDV-2 gB primers and MDV-3 gB primers were used, because I did not see any mention of MDV-2 gB PCR or MDV-3 (HVT) PCR in the Results.
(12)Table 2: The Ovo forward primer and Ovo reverse primer have been separated across two pages.
(13)Table 4: What is the significance of the shading in some rows?
(14)Line 307 (Immunofluorescence analysis section): The information about use of secondary antibodies seems to have been missed from this section (only primary Abs and counterstaining with DAPI are mentioned). Which secondary Abs (from the section above) were used to bind each primary Ab?
(15)Line 391: Change ‘particles’ to ‘DNA’ (because PCR detects the viral DNA, not viral particles).
(16)Line 384 (Molecular analysis section): I am confused by the mention of detection of meq in 75% of samples on line 387, but detection of meq in 85.7% of samples on line 391. Does the 75% refer to samples that were positive for all three tested genes (gB, gL and meq), and some further samples were positive for meq but negative for gB and gL?
(17)Table 8: I don’t understand how the superscript letters in the table relate to the key to superscript letters in the bottom row (for example, in the table, superscript ‘a’ is against MDV strain C12/130, but the key in the bottom row puts superscript ‘a’ against the MDV strain ATE2539).
(18)Line 558: Change ‘descendent’ to ‘descendant’.
(19)Figure 6: A clearer title would be ‘Effects of Nigeran MDV Meq isoform on cellular proliferation’.
(20)Line 737: Change ‘Nigerian’ to ‘Nigeria’.
(21)Line 845: Change ‘INF’ to ‘IFN’.
(22)Line 912: Change ‘exasperated’ to ‘exacerbated’.
Comments on the Quality of English LanguageUse of the English language is very good; there are a few words that seem incorrect and I have suggested corrections in the above section.
Reviewer 2 Report
Comments and Suggestions for Authors
This study assessed the field strains of Marek’s disease virus isolated in Nigeria. This work provides valuable insights into the possibility of the MDV evolution caused by vaccine pressures. However, there are some concerns regarding data analysis and reliability.
Major points:
1. The abstract lacks the background of MDV in Nigeria and the research objectives.
2. The materials and methods need improvements with clarification and further explanation as follows:
MDV genome copy number analysis: how did the authors figure out the number of the cells in FTA cards? Please add the details.
Antibodies: anti-rabbit IgG? Anti-mouse IgG? IgM?
CFSE MFI: gating strategy? single cell analysis? dead cell exclusion method?
Statistical analysis: The two-way ANOVA test is not an appropriate method to analyze the data. The test information of normality/equality of variances should be included.
Glycoprotein L mutation assay: The information of CEF should be included. The authors should use a uniformed strain name in the whole manuscript and figures (TK or TK-1 or TK1a?).
The materials and methods regarding Supplemental Table 2 and Supplemental Figure 2 should be described in the materials and methods section.
3. The descriptions in line 561-563 and line 598-600 are not supported clearly enough by the phylogenetic trees due to the low bootstrap values.
4. Canonical Nigerian Meq sequences should be added in Figure 4c and 5c.
5. The whole manuscript needs to be reconstructed.
Line 424-426, 451-454, 461-463, 494-496, 505-507, and 602-608 in the results section should be described in the discussion section.
Line 668-750 should be described in the introduction section.
Line 867-873 should be described in the results section.
Minor points:
1. Line 98: between 2015 and 2016
2. Line 139: The incidence and mortality rates should be included in Table 1.
3. As the pathotyping assay has not been done for Nigerian strain, the virulence of LEC-LG should be described as ND not vv++ in the Table 4, Figure 4, and Figure 5.
4. Line 490-494: The authors should revise this sentence. The percentage should be matched with Table 8.
5. Line 502: The authors should specify if GDA means GDA-FT or GDA-VB as they showed different identities in Table 7 and 8.
6. The annotations in Table 8 needs more explanations.
7. The resolution of Table 9 and Figure 1-5 should be improved.
8. It is not clear at which time point the data showed significant differences in Figure 6. The p value should be placed appropriately.

Some abbreviations need to be spelled out.
Round 2
Reviewer 2 Report
Comments and Suggestions for Authors
1. Dunnett's multiple comparison is for comparing each treatment/experimental group against a control group. Therefore, the authors cannot compare the LEC-LG Meq to the vvMDV RB1B Meq (Line 587-589). The authors should use a different multiple comparison test such as Tukey's multiple comparison test to compare all groups (if the data are parametric). The authors should include the test information of normality/equality of variances.
2. Discrimination of dead cells and duplet cells is highly recommended to acquire accurate data (Cossarizza et al. 2021). The authors should be able to use FSC-area vs FSC-width to gate single cells using FACSCalibur. As there are fixable dyes available for dead cell staining prior fixing cells, the description of “As all cells for the growth curve analysis were fixed, viable cell staining could not be used to separate live from dead cells.” (Line 336-337) is not appropriate.
Cossarizza, Andrea, Hyun‐Dong Chang, Andreas Radbruch, Sergio Abrignani, Richard Addo, Mübeccel Akdis, Immanuel Andrä, et al. 2021. “Guidelines for the Use of Flow Cytometry and Cell Sorting in Immunological Studies (Third Edition).” European Journal of Immunology 51 (12): 2708–3145. https://doi.org/10.1002/eji.202170126.
3. The authors should describe the bootstrap value cut off and reconsider the descriptions “NGH and BBL cluster in a clade of North American (RB1B, Md5, 584A, N, and 648A) strains” (Line 541-542) and “The PRR-CTD phylogenetic tree shows that the CTD among all high-virulence USA strains was closely related, having evolved differently from all low-virulence strains (Figure 5b)” (Line 573-575).
4. The authors should add the antibody information in the ‘Antibodies’ section same as in the Immunofluorescence analysis (IFA) section.
5. The modifications by the authors have not been reflected in the manuscript in line 156 and line 472-475. In addition, the authors should modify the percentages to “96.76 to 98.82 %” in line 472.
Comments on the Quality of English LanguageLine 32: Please remove parentheses, (12) farms to 12 farms.
